# Beyond Explicit Edges: Robust Reasoning over Noisy and Sparse Knowledge Graphs

**Hang Gao** [1]  **Dimitris N. Metaxas** [1]

## Abstract

GraphRAG is increasingly adopted for converting unstructured corpora into graph structures to enable multi-hop reasoning. However, standard graph algorithms rely heavily on static connectivity and explicit edges, often failing in real-world scenarios where Knowledge Graphs (KGs) are noisy, sparse, or incomplete. To address this limitation, we introduce INSES (Intelligent Navigation and Similarity Enhanced Search), a dynamic framework designed to reason beyond explicit edges. INSES couples LLM-guided navigation, which prunes noise and steers exploration, with embedding-based similarity expansion to recover hidden links and bridge semantic gaps. Recognizing the computational cost of graph reasoning, we complement INSES with a lightweight router that delegates simple queries to Naïve RAG and escalates complex cases to INSES, balancing efficiency with reasoning depth. Experimental results show that INSES performs favorably compared to established RAG and GraphRAG baselines on multiple benchmarks. In particular, on the MINE benchmark, it exhibits notable robustness and adaptability across KGs constructed by varying methods. Our code and data are publicly available at https://github.com/hanggao-gh/INSES.

## 1. Introduction

Graph search is a fundamental problem in computer science, serving as the engine for knowledge-graph reasoning, social network analysis, and bioinformatics. Classical algorithms such as Depth First Search, Breadth First Search, and Random Walk are typically adapted to knowledge graphs (KG) by operating over entity–relation triples. Although effective in idealized settings, these approaches face a fundamental mismatch when applied to real-world scenarios: they rely on *static connectivity* and *explicit edges*. However, real-world KGs are inherently noisy, sparse, and incomplete, containing semantic gaps that rigid traversal strategies cannot bridge. In these settings, KGs (Hogan et al., 2021; Ji et al., 2021; Paulheim, 2017) and social networks (Newman, 2003; Easley et al., 2010) are not just topological structures, but rich semantic repositories containing attributes and embedding representations. This opens a new opportunity: instead of passively traversing a flawed graph, can we utilize embeddings and Large Language Models (LLM) (Team et al., 2023; Dubey et al., 2024; OpenAI, 2024; ZhipuAI, 2024) to actively *navigate* and *repair* the structure during the search process?

Recent GraphRAG pipelines organize corpora into graph representations to enable multi-hop reasoning (Saxena et al., 2020; Procko & Ochoa, 2024; Hu et al., 2025), and LLM-guided / PPR variants have been proposed to refine traversal (Sun et al., 2024; Ma et al., 2025; Jimenez Gutierrez et al., 2024). However, exploration in most systems remains governed by explicit edges, which privilege local connectivity and ignore latent links. This is problematic because errors, redundancies, and missing links are unavoidable when extracting structured knowledge from natural language. Even with advanced construction methods such as OpenIE (Angeli et al., 2015), GraphRAG (Edge et al., 2024), or the recent KGGEN (Mo et al., 2025), imperfections persist, leading to fragmented relationships between semantically identical but distinct entities.

To illustrate this *semantic fragmentation*, consider an article titled "The Life Cycle of a Butterfly" in the MINE benchmark (Mo et al., 2025). Table 1 shows entity nodes generated by KGGEN, GraphRAG, and OpenIE. We observe a proliferation of similar yet disconnected entities, such as "butterflies," "adult butterflies," and "female butterflies". Merging these nodes risks information loss (semantic drift), while keeping them separate creates a sparse graph where reasoning trails are broken. Traditional search algorithms, bound by explicit edges, cannot jump from "butter-

---

[1]Department of Computer Science, Rutgers University, Piscataway, New Jersey, USA. Correspondence to: Hang Gao <h.gao@rutgers.edu>, Dimitris N. Metaxas <dnm@cs.rutgers.edu>.

*Proceedings of the 43rd International Conference on Machine Learning*, Seoul, South Korea. PMLR 306, 2026. Copyright 2026 by the author(s).

*Table 1.* Entity Nodes Generated by Different Methods

| Method | Entity Nodes Generated by Different Methods |
|---|---|
| KGGEN | ["adult", "adulthood", "antennae", "appearance", "appreciation", "balance", "beauty", "biodiversity", "birds", "body", "butterfly", "camouflage", "caterpillar", ..., **"food", "food source"**, "host plants", **"life cycle", "lifespan", "plant populations", "plants"**, ...] |
| GraphRAG | ["egg stage", "birds", **"adult butterflies"**, "nectar", "insects", "pupa", "host plants", "larva stage", "chrysalis", "metamorphosis", **"female butterflies"**, "pollination", "reptiles", **"butterflies"**, "caterpillar", **"butterfly"**, ...] |
| OpenIE | ["They", "egg to larva", "specific host plants", "journey filled", "third stage", **"butterfly 's life cycle", "lifespan ranging from few days to weeks", "Life Cycle"**, "changes", "laid", "twigs", "Next time", "to prepare for stage of its life cycle", ..., **"short lifespan ranging", "lifespan ranging from days", "prepare for stage of its life cycle", "lifespan ranging from days to weeks", "life cycle"**, ...] |

fly" to "adult butterflies" if no direct edge exists. However, these nodes share high embedding similarity. This suggests that the graph structure should not be treated as immutable; rather, latent semantic connections should be dynamically exploited to bridge gaps beyond explicit edges. By incorporating similarity-based expansion, we can turn the search process into an on-the-fly graph repair mechanism, surfacing hidden links needed for reasoning.

Building upon these insights, we propose INSES (Intelligent Navigation and Similarity-Enhanced Search), a robust framework that transforms graph traversal from a static walk into a dynamic, semantics-aware reasoning process. INSES addresses the dual challenges of noise and sparsity through two coupled mechanisms. First, to handle noise and reduce search space, an LLM navigator actively prunes adjacent triples, steering exploration toward query-relevant evidence. Second, to mitigate sparsity and fix broken paths, embedding-based similarity expansion dynamically augments the frontier with semantically proximate nodes. This effectively creates virtual edges during inference, recovering latent links missed by the construction phase. These components act in tandem: navigation prunes, while similarity connects/repairs. Furthermore, recognizing that not all queries require complex graph reasoning, we introduce a lightweight router. This module dispatches straightforward queries to Naïve RAG and escalates complex, multi-hop cases to INSES, thereby balancing computational cost with reasoning depth.

We evaluate INSES on three multi-hop QA benchmarks, observing consistent gains over strong RAG and GraphRAG baselines. The results demonstrate INSES's robustness to dataset difficulty and reasoning depth. Ablation studies confirm that similarity-based expansion is the dominant contributor to accuracy, validating our hypothesis on the

importance of recovering latent links, while the router effectively contains costs. Analysis shows that the router efficiently handles shallow queries, reserving the heavy lifting of INSES for complex cases. Finally, on the MINE benchmark, INSES demonstrates superior adaptability across KGs built by different methods. We summarize our main contributions as follows:

- We diagnose the limitation of explicit-edge exploration in noisy/incomplete KGs, identifying semantic fragmentation as a key bottleneck that requires dynamic, semantics-aware correction.

- We introduce INSES, a framework that fuses LLM-guided navigation with similarity-based expansion, enabling robust reasoning beyond explicit graph structures.

- We design a lightweight router that optimizes the accuracy-cost trade-off by preserving RAG-level efficiency for easy queries while escalating complex cases or low-confidence to INSES.

**Conflict of Interest Disclosure.** The authors declare that this research was conducted in the absence of any commercial or financial relationships that could be construed as a potential conflict of interest.

## 2. Related Work

### 2.1. Knowledge Graph Reasoning

Reasoning in knowledge graphs traditionally adapts search procedures (e.g. depth / breadth-oriented traversals, random walk) to operate over entity–relation triples rather than using general-graph routines verbatim, which implicitly assumes

that explicit edges are sufficient evidence trails (Wang et al., 2013; Yih et al., 2015; Sun et al., 2018; Lao & Cohen, 2010; Ristoski & Paulheim, 2016). Recent graph-centric pipelines construct or reorganize structure and then guide exploration for multi-hop reasoning. Neural Path Hunter (Dziri et al., 2021) utilizes KGs to verify and correct factual hallucinations in LLM responses by finding a supporting path in the KG for a given dialogue. Some form hierarchical/summary trees to route queries across levels (Sarthi et al., 2024; Zhang et al., 2025); others induce community-structured subgraphs for summary-centric retrieval (Edge et al., 2024); a third line dynamically constructs KGs and designs adaptive traversal policies (Li et al., 2024; Wang et al., 2024); further variants couple traversal with LLM decision-making (e.g., beam-style selection) (Sun et al., 2024; Ma et al., 2025) or employ importance-biased walks for multi-hop retrieval (Gutiérrez et al., 2024). Despite these advances, exploration is still largely governed by explicit connectivity and fixed local budgets, which under-captures cross-entity evidence and overlooks latent semantic relations (e.g., aliasing among similar-but-distinct nodes) that are not realized as direct triples. To move beyond edge-only locality and reduce noise from imperfect structure, INSES integrates LLM-guided navigation (pruning/steering triple selection using attributes and semantics) with embedding-based similarity expansion (temporarily extending the frontier with semantically proximate nodes to recover hidden links), turning structural walks into semantics-aware multi-hop reasoning under bounded iterations.

## 2.2. Retrieval Augmented Generation

Retrieval Augmented Generation (RAG) integrates retrieval into generation to ground LLMs in external knowledge, evolving from early retrieval-based QA (Chen et al., 2017; Karpukhin et al., 2020; Guu et al., 2020) to end-to-end coupling of retrieval and generation (Lewis et al., 2020), with recent advances using LLMs as retrievers (Yu et al., 2023; Sun et al., 2023) and finer retrieval granularity such as propositions (Chen et al., 2024). In practice, RAG spans text-based and KG-based: text-based variants retrieve semantically similar passages (Gao et al., 2023b; Zhao et al., 2024; Xiao et al., 2025; Chen et al., 2025) but can miss deeper relational structure and include redundant context; iterative schemes that interleave retrieval and reasoning (Shao et al., 2023; Trivedi et al., 2023; Wei et al., 2022; Gao et al., 2023a) improve recall but increase latency and risk error accumulation without a reliable guide. Graph-based RAG offers more interpretable and precise structure (Wang et al., 2024; Liang et al., 2025). Early work injected KG knowledge directly into model representations (Peters et al., 2019; Liu et al., 2020), while more recent approaches externally augment LLMs by translating relevant KG subgraphs into prompts (Wen et al., 2024; Dai et al., 2025; Zhang et al., 2024a),

while these pipelines inherit KG incompleteness. Together, these trade-offs motivate systems that preserve text-RAG efficiency in easy cases while invoking structured semantics-aware reasoning when needed. We address this tension with a lightweight router that keeps easy and shallow queries in standard RAG and escalates complex/low-confidence queries to INSES; once escalated, INSES's LLM navigation + similarity expansion directly targets static-connectivity blind spots by leveraging rich node attributes and embedding proximity during adaptive graph search.

## 3. Methodology

### 3.1. Dynamic Reasoning vs. Static Completion

Our framework is designed to function differently from traditional Knowledge Graph Completion (KGC) and Link Prediction (LP) techniques (Zhu et al., 2021; Wang et al., 2022; Galkin et al., 2024; Zhang et al., 2024b). Although these methods also leverage embedding similarity to infer missing links, they typically focus on offline static graph materialization. For complex reasoning tasks, such static densification can be detrimental. For instance, statically establishing a bidirectional equivalence between "butterflies" and "female butterflies" (example in Table 1) could erroneously propagate specific attributes (e.g., egg-laying behaviors) to the general category, introducing factual noise and semantic drift.

In contrast, our INSES employs similarity expansion dynamically during the query-specific search process. By coupling similarity expansion with LLM navigator, our framework utilizes the model's parametric knowledge to validate and prune these virtual edges on the fly. This ensures that latent semantic connections are exploited to bridge gaps without permanently corrupting the graph's precision with context-insensitive links. More importantly, the timely pruning by the LLM navigator effectively prevents the propagation of noise and semantic drift that similarity expansion might introduce.

We store KG as a property graph (Angles, 2018; Angles et al., 2017). Beyond basic node/edge connectivity, a property graph attaches rich attributes (e.g., textual descriptions, types) to nodes and edges, and each node further has an embedding representation. This allows search to exploit attribute filters and to fuse structural and embedding information, improving both efficiency and accuracy. We begin with formal definitions.

### 3.2. Preliminaries

**Definition 3.1.** (Property-Graph-based Knowledge Graph)

$$KG = (V, E, \lambda_V, \lambda_E, \phi),$$

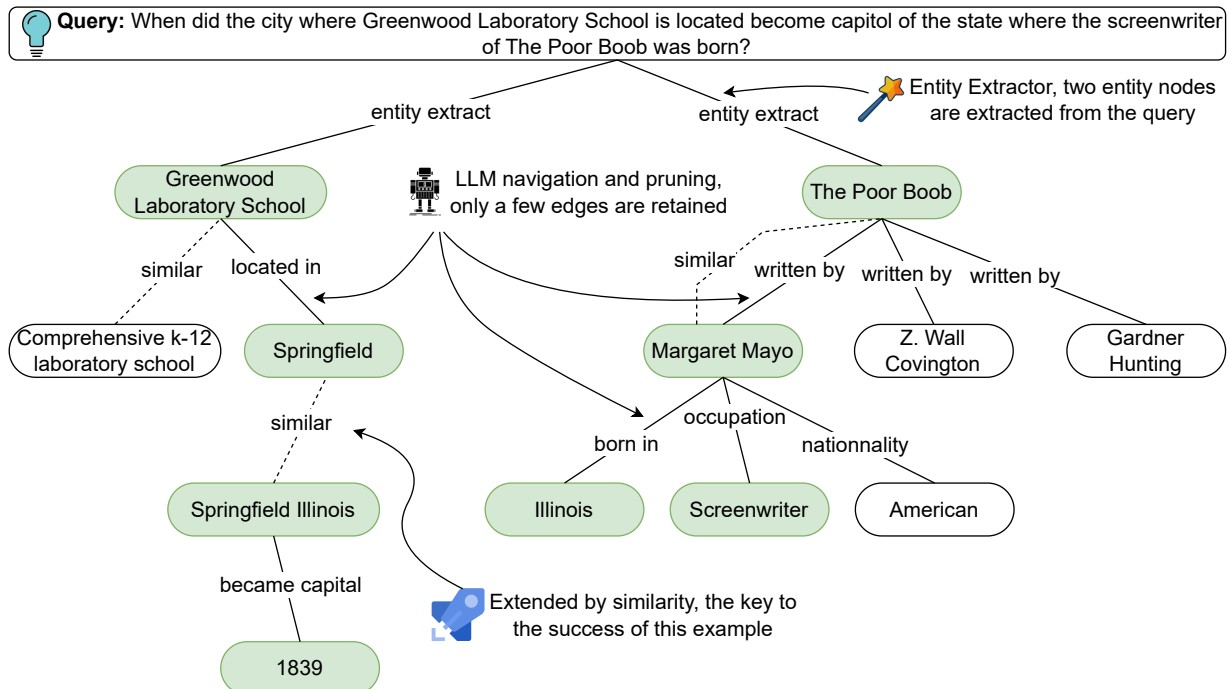

*Figure 1.* An example of INSES workflow. Solid edges denote explicit relations; dashed edges denote dynamically added similarity edges. Nodes/edges with green background aid answering query. LLM maps query entities ("Greenwood Laboratory School" and "The Poor Boob") to initial nodes, picks relevant triples while pruning noise, and uses similarity expansion to recover latent links (e.g., "Springfield" → "Springfield Illinois"). Navigation and pruning discard spurious edges, while expansion reveals critical connections, together enabling more reliable multi-hop reasoning.

where $V$ is the set of entity nodes; $E \subseteq V \times V$ is the set of semantic relation edges; $\lambda_V$ and $\lambda_E$ are attribute functions for nodes and edges; and $\phi : V \to \mathbb{R}^d$ maps each node to a $d$-dimensional embedding. For each edge $e = (u, v) \in E$, the corresponding knowledge triple is $(u, \lambda_E(e), v)$.

**Definition 3.2.** (Multi-hop Search on Knowledge Graphs) Given a natural-language query $q$, multi-hop search (reasoning) aims to identify triples in the graph that are relevant to $q$ and useful for answering it:

$$\mathcal{T}(q) = \{(u, \lambda_E(e), v) \in \mathcal{G} \mid$$
$$\text{Relevant}((u, \lambda_E(e), v), q) = \text{True}\},$$

where $\mathcal{G} = \{(u, \lambda_E(e), v) \mid (u, v) \in E\}$ is the set of all triples, $\mathcal{T}(q)$ denotes the evidence triples for $q$, and $\text{Relevant}(\cdot, q)$ is a relevance function.

Traditional graph search can be applied to this task, but exhaustive traversal on large KGs is neither computationally feasible nor necessary. Advances in LLM, embeddings, and graph representation learning enable a more intelligent, dynamic search. Node embeddings let us map nodes to a vector space and expand the graph via similarity; LLMs provide semantic-aware guidance to steer search and prune noise inherent in the graph.

**High-level workflow**. INSES first matches the query to semantically similar entity nodes via vector embeddings. It then iterates: at each step, (i) LLM selects informative triples from the neighbors of the current nodes, triples that directly support answering $q$ or are promising for further exploration; (ii) a similarity module finds nodes most similar to the current nodes. The LLM-selected neighbors and similarity-based nodes are merged to form the next current nodes. These steps repeat until the answer is found or the iteration limits are reached. Figure 1 shows an example workflow of INSES.

### 3.3. Step 1: Extract Initial Entity Nodes

Use an LLM to extract entities from $q$, that is:

$$\text{LLM}_{\text{Extractor}}(q) = \{m_i\}_{i=1}^k.$$

For each entity $m_i$, retrieve the entity node most similar in $KG$ by cosine similarity to form the initial node set:

$$V_{\text{init}} = \Big\{v_i \mid v_i = \arg\max_{v \in V} \cos\big(\phi(m_i), \phi(v)\big),$$
$$i = 1, \ldots, k\Big\}, \tag{1}$$

where $\phi(\cdot)$ denotes the embedding function consistent with the construction of $KG$.

### 3.4. Step 2: LLM Navigation

In this step, the adjacent triples of the current node, denoted by $T_{\text{adj}}$, are extracted and then pruned by LLM and judged whether they are sufficient to answer the question.

Initialize $V_{\text{current}} = V_{\text{init}}$, $T_{\text{selected}} = \varnothing$. Then

$$T_{\text{adj}} = \{(x, \lambda_E(e), y) \in \mathcal{G} \mid$$
$$e = (x, y) \in E, \ x \in V_{\text{current}} \text{ or } y \in V_{\text{current}}\}. \quad (2)$$

An LLM acts as a navigator, that is,

$$\text{LLM}_{\text{Navigator}}(q, T_{\text{selected}}, T_{\text{adj}}) \rightarrow$$
$$\begin{cases} \text{STOP}, & \text{if answerable;} \\ (T_{\text{new\_selected}}, V_{\text{candidate}}), & \text{otherwise,} \end{cases} \quad (3)$$

where $T_{\text{new\_selected}} \subseteq T_{\text{adj}}$ are newly selected triples and $V_{\text{candidate}}$ are endpoints of $T_{\text{new\_selected}}$.

### 3.5. Step 3: Similarity-based Expansion and Augmentation

Compute similar nodes for each $u \in V_{\text{current}}$ and keep those above a threshold $\tau_{\text{sim}}$:

$$V_{\text{sim}} = \{v \in V \setminus \{u\} \mid$$
$$v \in \text{Top-}k(u) \wedge \cos(\phi(u), \phi(v)) \geq \tau_{\text{sim}}\}, \quad (4)$$

where Top-$k(u)$ denotes the set of $k$ nodes in $V \setminus \{u\}$ that have the highest similarity scores with $u$.

Update $V_{current}$ by merging candidates and removing visited nodes:

$$V_{\text{current}} \leftarrow (V_{\text{candidate}} \cup V_{\text{sim}}) \setminus V_{\text{visited}}.$$

This dynamically augments structure beyond explicit edges to capture latent semantic links.

The complete algorithm is shown in Algorithm 1, and more details of implementation, as well as complexity and efficiency analysis, are provided in Appendix B and C.

### 3.6. Complexity Control and Routing

LLM-driven navigation introduces additional complexity, so we limit the number of navigation iterations to control cost, by default six, motivated by the theory of small world (Milgram et al., 1967; Kleinberg, 2000). Furthermore, previous work (Han et al., 2025; Zhou et al., 2025) and our experiments indicate that Graph-based RAG is not uniformly superior to Text RAG. They differ fundamentally at the structural level, naturally leading to distinct strengths and usage regimes. For most straightforward ($\leq 2$ hops) queries, Text RAG often suffices because the answer's immediate

---

**Algorithm 1** Intelligent Navigation and Similarity Enhanced Search (INSES)

---

**Input:** Property knowledge graph $KG = (V, E, \lambda_V, \lambda_E, \phi)$, query $q$ stated in natural language
**Output:** A set of triples $T_{\text{selected}}$ that are helpful in answering $q$

1: Use Equation 1 and query $q$ to establish the initial node set $V_{\text{init}}$
2: $V_{\text{visited}} = \{\}$
3: $T_{\text{selected}} = \{\}$
4: $V_{\text{current}} = V_{\text{init}}$
5: $iteration = 0$
6: **while** $iteration < max\_iter$ and $V_{\text{current}} \neq \varnothing$ **do**
7:     $V_{\text{visited}} = V_{\text{visited}} \cup V_{\text{current}}$
8:     Get adjacent triples $T_{\text{adj}}$ using Equation 2
9:     Use Equation 3 to let LLM select triples from $T_{\text{adj}}$, determine whether the current information is sufficient and return $T_{\text{new\_selected}}$ and $V_{\text{candidate}}$
10:     $T_{\text{selected}} = T_{\text{selected}} \cup T_{\text{new\_selected}}$
11:     **if** sufficient **then**
12:        break
13:     **end if**
14:     Use Equation 4 to find similar nodes $V_{\text{sim}}$
15:     $V_{\text{current}} = (V_{\text{candidate}} \cup V_{\text{sim}}) \setminus V_{\text{visited}}$
16:     $iteration = iteration + 1$
17: **end while**
18: return $T_{\text{selected}}$

---

neighborhood is typically explicitly mentioned or strongly evoked by type/semantic cues in the query. Conversely, for multi-hop ($\geq 3$) or long-range queries, explicit multi-hop connectivity in a KG becomes valuable to expose long-range correlations that Text RAG chunking might miss.

To balance quality and cost, we introduce a lightweight router that allows each method to specialize. Text RAG serves the abundant simple cases cheaply and with high fidelity to source wording, while INSES is reserved for genuinely complex problems where explicit structure is advantageous. Furthermore, considering the massive difference in computational cost (Graph-based RAG is usually several times slower than Text RAG due to multiple LLM calls and graph operations), a slightly conservative routing strategy can be adopted in real-world systems: All queries are initially routed to Text RAG, which provides an answer along with a confidence score. High-confidence answers are accepted directly, and only queries yielding low-confidence answers are subsequently routed to Graph-based RAG. This approach adds minimal overall cost while fully leveraging the respective strengths of both Text RAG and Graph-based RAG.

Our specific implementation, detailed in Algorithm 2, combines an initial complexity estimation with this confidence-

based fallback. An LLM first determines if a query requires multi-hop search. Simple queries are handled by a standard Naïve RAG pipeline with confidence estimation, whereas multi-hop queries or simple queries that yield low-confidence results from Naïve RAG are escalated to INSES for structured graph search and reasoning. Further granular analysis of Text RAG versus Graph-based RAG is provided in Appendix D.

---

**Algorithm 2** Router Algorithm

---

Input: A Naïve RAG system with a vector database, a knowledge graph $KG = (V, E, \lambda_V, \lambda_E, \phi)$, query $q$ stated in natural language

Output: A set of text or a set of triples that are helpful in answering $q$

1: Use LLM to determine if $q$ is related to multi-hop ($\geq 3$) search
2: **if** False **then**
3:    route to the Naïve RAG
4:    Naïve RAG gives an answer with $confidence$
5:    **if** $confidence > Confidence_{threshold}$ **then**
6:       return the results given by Naïve RAG
7:    **else**
8:       route to running INSES on $KG$
9:    **end if**
10: **else**
11:    route to running INSES on $KG$
12: **end if**
13: return the results given by INSES

---

# 4. Experiments

## 4.1. Datasets

To assess the effectiveness of INSES on graph search and reasoning, we conduct experiments on three widely used multi-hop benchmarks: MuSiQue (Trivedi et al., 2022), 2WikiMultiHopQA (Ho et al., 2020), and HotpotQA (Yang et al., 2018). For fairness, we follow the evaluation protocol of previous work such as IRCoT (Trivedi et al., 2023), ensuring that all methods retrieve from the same underlying corpus. To make the experiments computationally feasible while still representative, we sample 1,000 queries from each dataset as our test set.

## 4.2. Baselines

We compare our approach with three families of baselines. (i) LLM-only methods answer without external retrieval, including Direct Prompting (Direct), the model outputs the final answer without exemplars, and Few-shot CoT Prompting (Few-shot CoT) (Wei et al., 2022), where exemplars provide step-by-step rationales and final answers that the model emulates. (ii) Text-based RAG methods retrieve unstruc-

tured text and condition the LLM on retrieved snippets; we include the Naïve RAG pipeline, HyDE (Gao et al., 2023a) (which generates a hypothetical document from the query to guide retrieval) and IRCoT (Trivedi et al., 2023) (which interleaves iterative retrieval with chain-of-thought prompting). (iii) Graph-based RAG methods retrieve and reason over structured representations; we evaluate GraphRAG (Edge et al., 2024), LightRAG (Guo et al., 2024), RAPTOR (Sarthi et al., 2024), and SiReRAG (Zhang et al., 2025), which leverage graph/cluster structure to aggregate evidence for multi-hop reasoning.

## 4.3. Metrics

We evaluate all methods using two complementary metrics.

**Exact Match (EM).** EM measures whether the predicted answer string exactly matches the ground truth. This is a strict evaluation criterion that rewards only verbatim matches. While widely used in QA benchmarks, EM often underestimates performance when semantically correct answers differ slightly in surface form.

**LLM-as-a-Judge (LLM Judge).** To better capture semantic correctness, we adopt an evaluation protocol in which a LLM acts as a judge. Given the query $q$, the ground truth answer, and the model prediction, the LLM judge determines whether the prediction is semantically consistent with the ground truth and can be considered a correct answer to $q$. This approach mitigates the limitations of surface-level overlap and has recently been shown to be reliable and closely aligned with human evaluation in multiple studies covering various generative tasks (Gu et al., 2024).

## 4.4. Implementation Details

We follow a standard pipeline for constructing KGs from QA datasets. The constructed KG is stored in the Neo4j graph database (Robinson et al., 2015; Francis et al., 2018). For system integration, we adopt LlamaIndex (Liu, 2022), which offers a modular interface to connect LLMs, databases, and retrieval components in a unified framework. For the embedding model, we use the lightweight model bge-base-en-v1.5 (BAAI, 2024), chosen for its balance between precision and efficiency. Unless otherwise specified, all experiments use GLM-4 (ZhipuAI, 2024) as the LLM backbone for reasoning, navigation, and answer generation. To comprehensively evaluate the robustness and adaptability of our approach, we also include ablation studies and comparisons with stronger models - GPT-4o (OpenAI, 2024).

## 4.5. Main Results and Analysis

Table 2 reports the performance of all baselines and our proposed method on three datasets, which can be summarized as follows:

*Table 2.* Performance comparison among baselines and INSES on three benchmark datasets in terms of EM and LLM Judge.

| | Baseline methods | Musique | | 2Wiki | | HotpotQA | |
|---|---|---|---|---|---|---|---|
| | | EM | LLM Judge | EM | LLM Judge | EM | LLM Judge |
| *LLM only* | GLM-4 (Direct) | 0.15 | 0.18 | 0.32 | 0.36 | 0.41 | 0.49 |
| | GLM-4 (Few-shot CoT) | 0.24 | 0.27 | 0.38 | 0.46 | 0.51 | 0.57 |
| *Text-based* | Naïve RAG (Top-5) | 0.31 | 0.29 | 0.39 | 0.43 | 0.62 | 0.71 |
| | Naïve RAG (Top-10) | 0.33 | 0.37 | 0.41 | 0.44 | 0.67 | 0.77 |
| | HyDE | 0.21 | 0.31 | 0.45 | 0.46 | 0.57 | 0.63 |
| | IRCoT | 0.25 | 0.42 | 0.38 | 0.43 | 0.37 | 0.48 |
| *Graph-based* | GraphRAG (Top-5) | 0.23 | 0.24 | 0.38 | 0.35 | 0.43 | 0.63 |
| | GraphRAG (Top-10) | 0.26 | 0.36 | 0.50 | 0.43 | 0.47 | 0.61 |
| | LightRAG | 0.38 | 0.42 | 0.58 | 0.58 | 0.67 | 0.77 |
| | RAPTOR | 0.32 | 0.35 | 0.52 | 0.47 | 0.68 | 0.70 |
| | SiReRAG | 0.44 | 0.43 | 0.48 | 0.53 | 0.61 | 0.75 |
| **Ours** | **INSES + Router** | **0.46** | **0.47** | **0.67** | **0.71** | **0.68** | **0.80** |

- Our proposed INSES + Router consistently outperforms all baselines on both EM and LLM Judge across all datasets. The strongest baseline, SiReRAG, approaches our scores on Musique but shows a clear gap on 2Wiki and a non-trivial gap on HotpotQA.

- Several Graph-based variants (e.g., GraphRAG) fall short of Text RAG in these short, independent QA tasks. One reason is their reliance on cluster/community summaries as the basis for generation, an approach better suited to long, thematically related document sets than to brief factoid questions. In addition, as noted in the Introduction, there is no perfect procedure for text to KG conversion: real KGs are inevitably incomplete and noisy (missing/ambiguous links). Together with the granularity/organization mismatch, these factors imply different applicability regimes rather than an across-the-board advantage for graph methods. This motivates our routing design: since Text RAG is much cheaper than Graph-based pipelines, routing between Text RAG and INSES balances both performance and cost.

- Most Text RAG baselines are relatively stable, and several perform strongly on HotpotQA; for example, Naïve RAG (Top-10) comes close to our method on that dataset. This supports the view that Text RAG excels on simpler or short-chain questions.

- In most cases, LLM Judge and EM track closely. Larger gaps occur primarily on HotpotQA, suggesting that its answer format affects exact string matching more than semantic consistency, making LLM Judge a useful complementary metric there.

The results of the experiment highlight the adaptability and robustness of our approach and illustrate the complementary strengths of Text RAG and Graph-based RAG. Text RAG operates over relatively coarse-grained units (e.g., text chunks) with lower construction and retrieval costs, while Graph-based methods operate at the finer granularity of triples, leading to higher construction and retrieval overhead, but greater reasoning precision. These results also validate the design of our router mechanism: simple queries can be efficiently handled by Naïve RAG, while more complex multi-hop reasoning queries benefit from the Graph-based search of INSES.

### 4.6. Ablation Study

To better understand the contribution of each component in INSES, we perform a step-wise ablation study. Specifically, we evaluate the following settings: **(i)** using only the LLM Navigator; **(ii)** adding Similarity Enhancement on top of the LLM Navigator; and **(iii)** further incorporating the Router. All three variants employ GLM-4 as the underlying LLM. In addition, we test GPT-4o as a stronger backbone to examine the sensitivity of INSES to the choice of LLM.

Table 3 shows that the similarity expansion makes the largest contribution, resulting in substantial improvements of 0.12 (EM) on MuSiQue, 0.06 (EM) on 2Wiki, and 0.05(EM) on HotpotQA. These gains are more pronounced on complex queries, while simpler queries (often ≤2-hop) benefit less since multi-hop reasoning is not required. The router provides additional improvements, though smaller than those brought about by the similarity expansion. Switching from GLM-4 to GPT-4o leads to only modest gains, suggesting that navigation and similarity enhancement themselves are dominant factors; once the LLM is sufficiently competent, stronger backbones deliver diminishing returns.

Finally, the HotpotQA results reveal a key insight: Naïve

*Table 3.* Ablation study on three datasets.

| INSES | Musique | | 2Wiki | | HotpotQA | |
|---|---|---|---|---|---|---|
| | EM | LLM Judge | EM | LLM Judge | EM | LLM Judge |
| GLM-4 (Direct) | 0.15 | 0.18 | 0.32 | 0.36 | 0.41 | 0.49 |
| GPT-4o (Direct) | 0.28 | 0.35 | 0.54 | 0.57 | 0.49 | 0.65 |
| Naïve RAG (Top-5) | 0.31 | 0.29 | 0.39 | 0.43 | 0.62 | 0.71 |
| w/ LLM Navigator | 0.32 | 0.35 | 0.57 | 0.51 | 0.53 | 0.62 |
| w/ LLM Navigator + Similarity Enhance | 0.44 | 0.45 | 0.63 | 0.61 | 0.58 | 0.69 |
| w/ LLM Navigator + Similarity Enhance + Router | 0.46 | 0.47 | 0.67 | 0.71 | 0.68 | 0.80 |
| w/ LLM Navigator + Similarity Enhance + Router (GPT-4o) | 0.48 | 0.49 | 0.69 | 0.73 | 0.68 | 0.79 |

RAG already performs well on simpler cases, sometimes outperforming graph search, which highlights the router's particular value. By assigning straightforward queries to Naïve RAG and applying INSES to complex reasoning tasks, the system strikes a balance between cost and performance.

## 4.7. Routing Behavior Analysis

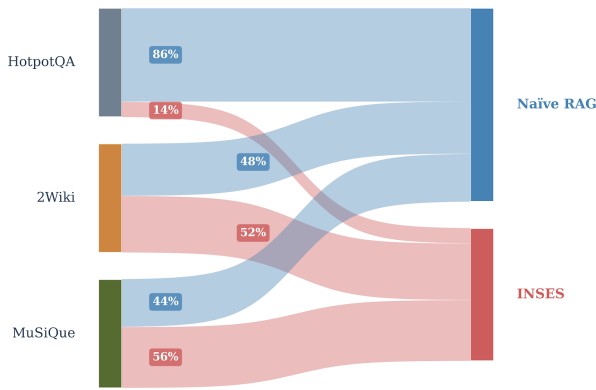

*Figure 2.* Proportion of queries routed to Naïve RAG vs. INSES across three datasets.

To better understand how the router balances efficiency and reasoning accuracy, we analyze the proportion of queries assigned to RAG versus INSES across datasets. This analysis reveals the practical role of the router: Whether it effectively delegates simple queries to lightweight retrieval while reserving graph-based reasoning for complex cases.

Figure 2 reports the fraction of queries routed to Naïve RAG and INSES on each dataset. The high share of Naïve RAG on HotpotQA (86%) indicates that many validation queries can be solved by shallow retrieval; consequently, the marginal benefit of graph search is smaller in this dataset. In contrast, MuSiQue and 2Wiki show near-balanced routing. This observation aligns with our ablation results (Table 3), where similarity expansion and multi-hop search yield larger gains on more complex queries.

## 4.8. Robustness and Adaptability of INSES

To assess the robustness and adaptability of INSES in KGs of different structure and quality, we evaluate it on MINE benchmark introduced in KGGEN (Mo et al., 2025). MINE contains 100 articles (each article contains approximately 1,000 words) covering 100 diverse topics, including history, art, science, ethics, and psychology. Each article is associated with 15 factual statements that are directly grounded in the content of the source article.

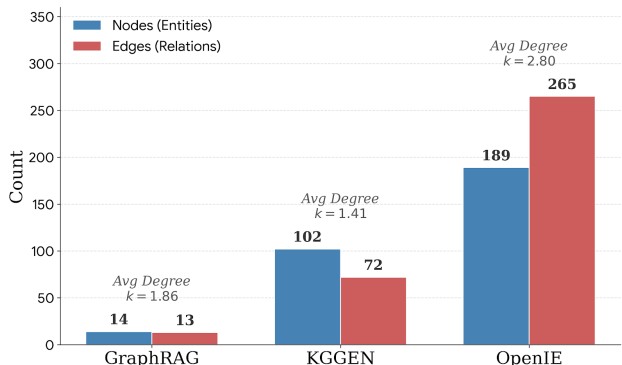

*Figure 3.* Quantitative comparison of knowledge graph topologies generated by GraphRAG, KGGEN, and OpenIE on the MINE dataset (averaged over 100 articles). The significant variance in node/edge counts and graph density underscores the heterogeneity of real-world KGs, necessitating the adaptive navigation and dynamic expansion capabilities of INSES to handle both sparsity and noise.

For each article, three KGs are generated by KGGEN, GraphRAG (Edge et al., 2024), and OpenIE (Angeli et al., 2015), representing distinct paradigms with varying density.

Figure 3 presents a comparative analysis of knowledge graphs constructed from the same corpus (MINE dataset (Mo et al., 2025)) using three distinct paradigms: GraphRAG, KGGEN, and OpenIE. Quantitative disparities are substantial. OpenIE generates the most extensive graphs (avg. 189 nodes, 265 edges), approximately $13.5\times$ more nodes and $20\times$ more edges than GraphRAG (avg. 14 nodes, 13 edges), with KGGEN positioned in between (avg. 102

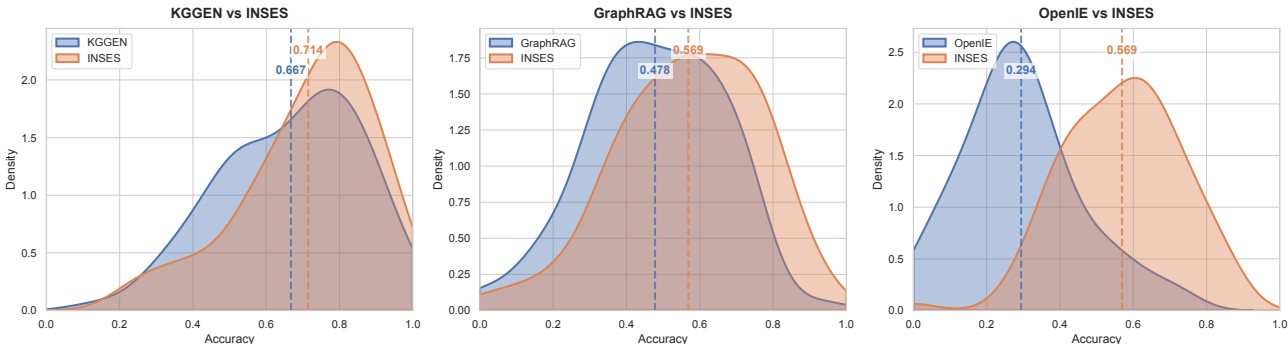

*Figure 4.* Kernel density distributions of query accuracy comparision across KGs constructed by three distinct methods. Dashed lines denote the mean accuracy for each distribution.

nodes, 72 edges).

Analyzing the graph density reveals a clear topological gradient: the average degree ranges from $1.41$ (KGGEN) and $1.86$ (GraphRAG) to $2.80$ (OpenIE). These discrepancies highlight a fundamental trade-off in extraction paradigms: compact graphs (like GraphRAG) minimize noise but risk severe *incompleteness* and *sparsity*, whereas extensive graphs (like OpenIE) capture more details but introduce significant *ambiguity* and *redundancy*.

This structural variability poses a dilemma for static search algorithms: a rigid traversal tailored for a dense graph may get lost in the noise of OpenIE, while the same algorithm may fail to find any path in the sparse structure of GraphRAG. Consequently, a robust reasoning framework must be adaptive. This validates the dual-mechanism design of INSES: *LLM-guided navigation* provides the necessary semantic filtering to handle the noise in dense graphs, while *similarity expansion* dynamically recovers latent links to bridge the gaps in sparse or incomplete graphs.

To empirically verify this structural adaptability in practice, we perform a comparative evaluation. First, we use the native retriever of each method to retrieve supporting triples for the 15 factual statements. An LLM then judges whether the retrieved triples are sufficient to infer the target fact; a query is scored 1 if sufficient (correct), otherwise 0. The accuracy per article is the number of correct queries divided by 15, and we report accuracies across all 100 articles. For comparison, INSES is also run on each of the three KGs, with the same evaluation procedure.

Figure 4 visualizes the kernel density distributions of query accuracy for INSES compared to the native retrievers of KGGEN, GraphRAG and OpenIE across 100 articles. In particular, KGGEN, GraphRAG, and OpenIE represent distinct construction paradigms that produce knowledge graphs with markedly different structures, sparsity, and noise levels. Despite this structural heterogeneity, INSES exhibits notable robustness: it consistently shifts the density mass toward higher accuracy segments. Quantitatively, INSES produces mean accuracy improvements of 4.7%, 9.1%, and 27.5% over the respective baselines. This stable adaptability validates the core design of INSES, demonstrating that the effective integration of dynamic similarity expansion and robust LLM-guided navigation can successfully mitigate the negative impacts of structural noise and incomplete explicit edges.

## 5. Conclusion

In this work, we present INSES, a multi-hop reasoning framework that transforms graph traversal from a rigid structural walk often constrained by explicit connectivity into a dynamic, semantics-aware process. By coupling LLM-guided navigation with embedding-based similarity expansion, INSES effectively mitigates the sparsity and noise inherent in real-world knowledge graphs, enabling robust inference even when critical structural links are missing.

A pivotal contribution of our work is the strategic shift from static graph completion to dynamic query-specific expansion. Unlike traditional knowledge graph completion or link prediction methods, which indiscriminately materialize missing edges offline, potentially introducing noise that harms general purpose reasoning, INSES keeps the underlying graph structure clean. Instead, it creates virtual edges on the fly, allowing the system to discover unconventional but contextually valid connections. The coupled LLM navigator ensures that these dynamic similarity expansions are strictly controlled and relevant, preventing noise propagation and semantic drift. Beyond specific KG reasoning tasks, INSES offers a broader methodological insight: it serves as a semantic extension to classical graph search algorithms such as DFS, BFS, and Random Walk. By integrating embedding-based dynamic expansion, INSES liberates graph search from the constraints of explicit connectivity, providing a new perspective for graph traversal tasks in scenarios with incomplete or implicit structure.

## Impact Statement

This paper presents work whose goal is to advance knowledge graph reasoning and retrieval-augmented generation. By introducing INSES, we enable robust reasoning over noisy and sparse graphs. This benefits downstream applications like automated fact-checking and scientific analysis, while our routing mechanism promotes computationally efficient and sustainable AI. However, because our framework relies on LLMs and embeddings to dynamically infer missing links, it risks propagating encoded biases and generating spurious connections. To mitigate this, INSES inherently employs an LLM-guided navigator to strictly verify and prune noisy semantic links. We urge researchers to pair such dynamic reasoning systems with rigorous fairness evaluations and human oversight in high-stakes domains.

## Acknowledgments

This research has been partially funded by research grants to Dimitris N. Metaxas through NSF: 2310966, 2235405, 2212301, 2003874, FA9550-23-1-0417, and NIH-5R01HL127661.

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

# APPENDIX

## A. Limitations

Although the proposed INSES framework significantly improves reasoning performance on noisy and incomplete knowledge structures, it introduces certain computational trade-offs and inherits inherent challenges associated with LLMs. For intricate multi-hop queries that bypass the lightweight router and escalate to the core search loop, the algorithm relies on iterative, sequential LLM calls to guide navigation and prune candidates. Although the routing mechanism successfully filters out the vast majority of straightforward queries to contain aggregate costs, the multi-turn interaction required for complex cases inherently increases end-to-end latency and API token consumption compared to standard Text RAG pipelines. Another issue we need to consider is the inherent vulnerabilities of LLM integration. As an increasing number of systems, including our framework, rely on LLM APIs to execute complex tasks, they inevitably inherit shared vulnerabilities associated with LLM integration. Chief among these is the instability of APIs, which exposes deterministic system architectures to silent model updates . For instance, unannounced changes to API outputs, such as unexpectedly encapsulating requested JSON formats within Markdown blocks, frequently broke the strict parsing scripts and caused evaluation errors, revealing a fundamental tension between the stochastic nature of LLMs and traditional deterministic software logic. Furthermore, all such systems face the universal challenge of prompt brittleness, because LLMs serve as the cognitive engine for critical operations. These limitations underscore the need for future LLM alignment to focus more heavily on strict format constraints. Concurrently, it suggests that future intelligent algorithms and agentic workflows must be designed with enhanced fault-tolerance and adaptive parsing mechanisms to robustly handle the intrinsic non-determinism of LLMs.

## B. Implementation Details of INSES Algorithm

*Table 4.* LLM Extract Entities Prompt

| **LLM Extract Entities Prompt** |
| --- |
| Your task is to extract several entities from the given query, so they can be used to search a knowledge graph for clues relevant to answering the query.
Return only the entities you extract, separated by commas, with no other text.

Query: {query} |

This section details the prompt designs for the core components of INSES: entity extraction, graph navigation, and final reasoning. These prompts are critical for interfacing the LLM's parametric knowledge with the structured data of the KG.

**Step 1: Initial Entity Extraction.** As defined in Equation 1 of Algorithm 1, the first step requires grounding the natural language query into specific entry points within the graph. We employ an LLM to extract key entities from the query, which serve as anchors for the initial vector similarity search. The specific prompt is provided in Table 4.

*Table 5.* LLM Navigation Prompt

---

**LLM Navigation Prompt**

---

Your task is to provide support for complex queries and multi-hop reasoning in the knowledge graph. Based on the following query, the visited nodes and the selected triplets, as well as the current nodes and their adjacent triplets, select the triplet numbers (separated by commas) from the adjacent triplets of the current nodes that help answer the query.

Selection criteria: Select the triplets that are most relevant to the query and most likely to help answer it.

Then determine:

1. Based on the visited nodes, the selected triplets, and the triplets you just selected, is this information sufficient to answer the query?

2. If so, answer "sufficient";

3. If not, answer "insufficient";

Your response must be in JSON format with two fields:

"determination": "sufficient/insufficient",

"selection": "triplet numbers, e.g., 1, 2, 3"

Query: {query}

The visited nodes:

{chr(10).join(visited_nodes_info) if visited_nodes_info else 'none'}

The selected triplets and their corresponding source text:

{chr(10).join(all_selected_triplets_info) if all_selected_triplets_info else 'none'}

The current nodes:

{chr(10).join(current_nodes_info) if current_nodes_info else 'none'}

The adjacent triplets and their corresponding source text:

{chr(10).join(current_triplets_info) if current_triplets_info else 'none'}

---

**Step 2: LLM-Guided Navigation.**    The core of INSES is the iterative navigation process (Equation 3 in Algorithm 1). At each step, the LLM must evaluate candidate triples not just for semantic relevance, but also for their potential to advance the reasoning chain. As shown in Table 5, the prompt provides the LLM with the *visited path*, *current context*, and *adjacent candidates*, asking it to perform two tasks: (1) select the most promising next-hop triples, and (2) determine if sufficient information has been gathered to terminate the search.

**Step 3: Reasoning and Answering.**    Once the navigation terminates (either due to sufficiency or iteration limits), the collected evidence triples are fed into the LLM to generate the final answer. Table 6 presents the template used for this generation. We explicitly enforce a "Chain-of-Thought" (CoT) format within the JSON output ("reasoning" field) to ensure the answer is grounded in the retrieved graph structure rather than the model's internal hallucinations.

**Hyperparameter Settings.**    To ensure the reproducibility of our experiments and maintain stable performance, we establish the following hyperparameter settings across all implementations. First, the temperature for all LLM generations is strictly set to 0.0. This strategy minimizes generation variance and prevents the model from hallucinating reasoning paths outside the provided context. Second, during the similarity-based expansion phase, the cosine similarity threshold ($\tau_{\text{sim}}$) is set to 0.8. This threshold guaranties that only semantically proximate and highly relevant latent nodes are added to the search frontier, preventing excessive noise. Finally, within the routing mechanism, the confidence threshold ($Confidence_{threshold}$) for the Naïve RAG pipeline is also set to 0.8. This value optimally balances efficiency and reasoning depth by accepting high-confidence simple answers while escalating uncertain or complex queries to INSES graph search pipeline.

## C. Extended Algorithmic Analysis and Methodological Positioning

In this section, we provide a detailed analysis of the computational complexity of INSES and clarify its methodological distinction from previous work such as Link Prediction and ToG.

*Table 6.* LLM Answer Question With Retrieved Triples Prompt

---

**LLM Answer Question With Retrieved Triplets Prompt**

---

You are a helpful assistant that provides accurate and concise answers based on the provided knowledge graph information. Please answer the following query: {query}

The following information is extracted from a knowledge graph, which contains entities, relationships, and relevant text: {context}

Your response must be in JSON format with two fields:
1. "reasoning": Your step-by-step reasoning process based on the knowledge graph information. Explain how the entities and relationships help answer the query.
2. "answer": The final answer to the query, as concise as possible without unnecessary explanations.

Example response format:
{{
"reasoning": "Step 1: Identified entity X and its relationship to entity Y. Step 2: Found that entity Z is connected to both X and Y. Step 3: Based on these relationships, concluded that...",
"answer": "Concise answer here"
}}
JSON Response:

---

## C.1. Computational Complexity Analysis

A concern regarding the efficiency of similarity-based expansion is the potential for quadratic complexity, i.e. $O(|V|^2)$, if pairwise comparisons were performed over the entire graph. However, the design of INSES avoids this through two key mechanisms: LLM-guided pruning and efficient vector indexing.

Referencing Algorithm 1, the complexity of a single iteration can be decomposed as follows:

1. **Bounded Search Frontier (LLM Pruning):** In the navigation step (Lines 8-9), the LLM acts as a semantic filter. Although the number of adjacent triples may be large, the LLM selects only a small, constant number of the most promising candidates ($T_{\text{new\_selected}}$) to expand. Consequently, the size of the current active node set, $|V_{\text{current}}|$, does not grow exponentially but remains bounded by a small constant $k$. Thus, the number of nodes requiring similarity checks in Line 14 is $O(1)$ relative to the total graph size $|V|$.

2. **Logarithmic Similarity Search:** The similarity-based expansion (Line 14) does not perform a brute-force scan. Instead, we leverage state-of-the-art Approximate Nearest Neighbor (ANN) search algorithms, specifically Hierarchical Navigable Small World (HNSW) graphs (Malkov & Yashunin, 2020; Malkov et al., 2014), which are standard in modern vector databases. The time complexity for retrieving the top-$k$ similar neighbors for a given node using HNSW is $O(\log |V|)$.

Therefore, the overall complexity for the similarity expansion step in INSES is approximately $O(k \cdot \log |V|)$, where $k$ is the negligible constant of active nodes. This logarithmic scaling ensures that INSES remains efficient even as the Knowledge Graph scales to millions of entities.

## C.2. Differentiation from Prior Paradigms

While INSES shares components with frameworks like Link Prediction (LP) and ToG, it represents a distinct algorithmic paradigm driven by the *tight coupling* of navigation and dynamic expansion.

**Comparison with Link Prediction (LP):** Traditional LP typically focuses on *static, offline* graph completion. The goal is to materialize missing edges globally before inference time. However, applying LP statically to reasoning tasks creates a fundamental dilemma: aggressive completion introduces massive amounts of noise (false positives) that derail

reasoning, while conservative completion fails to bridge semantic gaps. INSES resolves this via dynamic, query-specific expansion. We do not permanently add edges to the graph. Instead, *virtual edges* are generated on-the-fly only when relevant to the current query context. Crucially, these virtual edges are immediately subjected to verification by the LLM Navigator. If a similarity-based link is semantically irrelevant or erroneous, the LLM prunes it in the subsequent step. This generate-then-verify cycle allows INSES to utilize high-recall expansion without suffering from the precision drop typical of static LP.

**Comparison with Think-on-Graph (ToG):** ToG (Sun et al., 2024) relies on beam search on explicit graph structures. It is strictly bound by the topological connectivity of the KG. INSES transcends this by integrating the vector space into the topological space. By treating high-similarity pairs as latent connections, INSES can "teleport" across disconnected regions of the graph that are semantically contiguous. This is not merely an engineering integration but a shift from *structural traversal* to *semantic-structural co-navigation*.

### C.3. The Role of the Router

Finally, with respect to the routing mechanism, it is important to clarify that the router is not a remedy for algorithmic inefficiency but an architectural optimization for resource allocation. Regardless of the efficiency of the graph search algorithm itself, any graph-based reasoning pipeline that involves multiple LLM calls for neighborhood analysis is inherently more computationally expensive than Naïve RAG. Since a significant portion of real-world queries are easy and solvable by shallow retrieval, applying deep graph reasoning to them is resource-inefficient. The router ensures that the computational budget of INSES is invested solely in complex, multi-hop scenarios where deep reasoning yields tangible accuracy gains. This design aligns with the principle of adaptive computation rather than algorithmic patching. The next section provides a more detailed comparison between Text RAG and Graph-based RAG.

## D. Extended Comparison: Text RAG vs. Graph-based RAG

As discussed in Section 3.6, Text RAG and Graph-based RAG operate at different structural levels, naturally leading to distinct strengths and usage regimes. Table 7 systematically contrasts the two paradigms across data indexing, retrieval characteristics, generation trade-offs, and computational costs.

To further illustrate why Text RAG is highly effective for $\leq$2-hop queries, consider the following specific scenarios where the answer's immediate neighborhood is easily surfaced via dense similarity:

- **Converging 2-hop** ($A \rightarrow \text{Ans} \leftarrow C$): the answer node is directly adjacent to two entities named in the query. Chunks that mention the answer along with $A$ or $C$ are readily retrieved by dense similarity. For example:

  **Q:** Who authored *Pride and Prejudice* and was the sister of Cassandra Austen?
  **Reasoning:** *Pride and Prejudice* $\rightarrow$ JANE AUSTEN $\leftarrow$ Cassandra Austen.
  **Answer:** Jane Austen.

  Vector retrieval readily surfaces chunks where Jane Austen anchors both query mentions.

- **Chained 2-hop** ($A \rightarrow B \rightarrow \text{Ans}$): even if $B$ and Ans are not named, the query typically carries *type cues* that pull the right evidence in embedding space. Such type/entity clusters and relation patterns are well captured by modern embeddings, so relevant chunks co-mentioning $B$ and Ans are frequently surfaced. For example:

  **Q:** What river flows through the city that is home to the Eiffel Tower?
  **Reasoning:** Eiffel Tower $\rightarrow$ PARIS $\rightarrow$ SEINE.
  **Answer:** Seine.

  The answer is tightly tied to a query cue ("river"), which embeddings capture reliably.

Hence, with appropriate chunking, Text RAG handles many factual and $\leq$2-hop queries efficiently while preserving fine-grained details.

When the hop length exceeds these simple structures ($\geq$3 hops) or requires long-range reasoning, the probability of retrieving all necessary intermediate evidence via independent dense chunks drops significantly. In these instances, the

*Table 7.* Text RAG vs. Graph-based RAG: comparison of pipeline, capabilities, and costs

| Dimension | Text RAG | Graph-based RAG |
|---|---|---|
| Data form & indexing | Chunk raw text and retrieve via dense vectors; preserves original wording and details | Extract entities/relations or community structure to build a graph, then retrieve by graph and aggregate subgraphs/communities |
| Retrieval characteristics | Strong at in-place factual recall via semantic similarity; sensitive to type words and relation templates; well-suited to short-chain reasoning | Connects evidence across segments via explicit structure; better for long chains/hierarchical reasoning and thematic/context integration |
| Strengths (tasks) | Single-hop and $\leq$2-hop factual QA with rich details; pulling key snippets across documents | Multi-hop ($\geq$3) and long-range reasoning; contextual summarization/thematic synthesis; structured evidence integration |
| Generation trade-offs | Higher context relevance and lower noise; more focused coverage in creative/synthesis tasks | Broader evidence recall and coverage, but more redundancy; typical trade-off: coverage $\uparrow$ vs. relevance $\downarrow$ in creative tasks |
| Efficiency & cost | Low build/query cost; shorter prompts | Higher graph construction cost; retrieval/aggregation tends to inflate prompt length, increasing cost (varies by implementation) |
| Implementation variants | Classic dense retrieval with optional reranking, HyDE, hybrid (sparse+dense) retrieval | KG-style triple retrieval, community-based global/local retrieval, mixed nodes (concepts/passages), etc. |
| Common failure modes | Long-range/cross-document reasoning is hard; chunk boundaries hide global structure | Detail loss/missing or ambiguous links and noise during graph construction can cause retrieval drift; global summarization may lose fine details |
| Routing & hybrid use | Well-suited to handling factual/detail-oriented queries on its own | Well-suited to reasoning/multi-hop queries; can be integrated with or selectively routed alongside Text RAG for complementarity |

explicit structural chaining provided by a Knowledge Graph becomes essential, justifying the escalation to our INSES framework.

A router lets each method specialize. Text RAG serves the abundant simple cases cheaply and with high fidelity to source wording, while Graph-based RAG is reserved for genuinely multi-hop ($\geq$3) or long-range problems where explicit structure is advantageous. This not only balances quality and cost, but also improves robustness: when type cues suffice, dense retrieval excels; when structural chaining is essential, graph reasoning takes over. In our system, this routing criterion aligns with the structural characteristics above and reflects the empirical boundary between the two regimes. The specific router prompt is provided in Table 8.

## E. Case Study of INSES Algorithm

Table 9 is an example of INSES search without similarity expansion.

Table 10 is an example of INSES search with similarity expansion.

Below we analyze a concrete run of the INSES algorithm to illustrate how LLM navigation and similarity expansion work in practice. Table 9 shows the execution without similarity expansion, while Table10 shows the full INSES run with similarity expansion. Comparing the two, Table 9 fails to reach the correct answer yet demonstrates the effectiveness of LLM-based navigation; Table 10 succeeds, highlighting the effectiveness of similarity expansion and showing that during navigation the LLM not only selects relevant information but also filters out errors introduced by similarity expansion. A detailed analysis follows.

From Table 9, the initial entities extracted from the query "Who was the spouse of a leading speaker against slavery and publisher of an antislavery newspaper?" are four: ['leading speaker against slavery', 'antislavery newspaper',

*Table 8.* LLM Router Prompt

---

**LLM Router Prompt**

---

You are a strict binary classifier of question complexity for QA routing. Return EXACTLY one token: True or False. No punctuation. No explanation.

Definition:
- "True" ONLY if the question is exceptionally complex and strictly requires $\geq 3$ distinct reasoning hops.
- "False" if it can be answered with 1-2 hops, involves basic aggregations, or if you are unsure.
- DEFAULT RULE: When in doubt or if the complexity is borderline, strictly output False.

Examples:
Q: Who wrote 'Pride and Prejudice'?
A: False
Q: Capital of Spain?
A: False
Q: What is the currency of the country whose capital is Cairo?
A: False
Q: What is the capital of the country where New York is located? A: False
Q: Which city has a larger population, the capital of France or the capital of Italy?
A: False
Q: Which scientist discovered the element isolated by the spouse of the founder of X-ray crystallography?
A: True
Q: Find the museum that houses the painting created by the student of the artist who co-founded Cubism, and then report the city of that museum.
A: True

Q: {query}
A:

---

'spouse', 'publisher']. Using embedding similarity, these are matched to four nodes in the graph to form $V_{\text{init}}$: ['Opponent of slavery', 'Anti-slavery newspaper', 'Husband and wife', 'Newspaper publisher'].

In iteration 0, the LLM selects three triples from the neighborhood of these four entity nodes (recorded as $T_{\text{new\_selected}}$). This shows that the LLM navigates and prunes well, avoiding a flood of irrelevant triples. One reason is that rich attribute information in the property graph provides strong support for LLM decision-making; another is that, given the context, choosing relevant clues among available information is not a particularly hard task, so the LLM can keep the search breadth within a reasonable range.

In iteration 1, the LLM selects only one triple (again recorded in $T_{\text{new\_selected}}$): "The North Star $\rightarrow$ Published by $\rightarrow$ Frederick Douglass". Consequently, the candidate set for the next step is a single node, $V_{\text{candidate}} = [\text{'Frederick Douglass'}]$.

In iteration 2, from the neighborhood of node "Frederick Douglass" the LLM again selects only one triple—"The North Star $\rightarrow$ Published by $\rightarrow$ Frederick Douglass"—which had already been visited before. In other words, the opposite-end node of this triple has already been explored. No new candidate nodes are generated in this round, i.e., $V_{\text{candidate}} = \emptyset$. The search therefore terminates without finding the correct answer. Overall, the process shows that LLM navigation is efficient and does not select excessive irrelevant information.

Now consider the process in Table 10. Iterations 0, 1, and 2 proceed similarly to Table 9 but with similarity expansion applied at each round. The LLM's core selections remain essentially the same as in Table 9, and it promptly filters out errors introduced by similarity expansion. In Table 9, iteration 2 produces no new candidates and the search stops. In contrast, in Table 10's iteration 2, the current node "Frederick Douglass" yields a new node via similarity expansion—"Frederick Douglass Memorial and Historical Association"—and this newly surfaced node is precisely what leads to the final correct answer.

*Table 9.* Search without similarity expansion

| Iteration | The relevant status of each iteration |
|---|---|
| Query | Who was the spouse of a leading speaker against slavery and publisher of an antislavery newspaper? |
| Entities | ['leading speaker against slavery', *'antislavery newspaper'*, 'spouse', 'publisher'] |
| $V_{init}$ | ['Opponent of slavery', *'Anti-slavery newspaper'*, 'Husband and wife', 'Newspaper publisher'] |
| iter=0 | $V_{current}$: ['Opponent of slavery', *'Anti-slavery newspaper'*, 'Husband and wife', 'Newspaper publisher']. $T_{new\_selected}$: ['Thomas spottswood hinde → Occupation → Opponent of slavery', *'The north star → Is → Anti-slavery newspaper'*, 'Enos bronson → Was → Newspaper publisher']. $V_{candidate}$: ['Thomas spottswood hinde', *'The north star'*, 'Enos bronson'] |
| iter=1 | $V_{current}$: ['Thomas spottswood hinde', *'The north star'*, 'Enos bronson']. $T_{new\_selected}$: [*'The north star → Published by → Frederick douglass'*]. $V_{candidate}$: [*'Frederick douglass'*] |
| iter=2 | $V_{current}$: [*'Frederick douglass'*]. $T_{new\_selected}$: [*'The north star → Published by → Frederick douglass'*]. $V_{candidate}$: [ ] |
| Answer | Not Found. |

In iteration 3, the LLM selects the triple "Helen Pitts Douglass → Created → Frederick Douglass Memorial and Historical Association," whose opposite-end node is "Helen Pitts Douglass." In iteration 4, the LLM again selects a single triple—"Helen Pitts Douglass → Is → Second wife of Frederick Douglass"—which directly points to the correct answer. Note that in iterations 3 and 4 the LLM selects very few triples (only one each time) and is not distracted by irrelevant nodes introduced through similarity expansion; instead, it filters them out in a timely manner. This demonstrates that combining LLM navigation with similarity expansion is highly effective: similarity expansion can surface latent links, while LLM navigation can promptly prune potential errors introduced by that expansion.

An additional observation is that the final triple "Helen Pitts Douglass → Is → Second wife of Frederick Douglass" implies that "Second wife of Frederick Douglass" is modeled as a node in the KG. This also explains why the process in Table 9 failed to find the correct answer: in the constructed KG, "Frederick Douglass" and "Second wife of Frederick Douglass" are two separate nodes. As noted in the Introduction, it is difficult to convert natural-language information into a perfect KG. In this example, the fact "Helen Pitts Douglass is the second wife of Frederick Douglass" can be represented either as "Helen Pitts Douglass → Is → Second wife of Frederick Douglass" or as "Helen Pitts Douglass → Is the second wife of → Frederick Douglass," and both representations are reasonable. Such situations are common in KGs. If search and reasoning over a KG rely only on exact structural links, potential connections may be missed. Introducing similarity expansion is therefore an effective way to mitigate ambiguity and incompleteness.

## F. Prompts for LLM-only Baselines

To ensure the reproducibility of our experiments and to rigorously benchmark the contribution of external knowledge retrieval, we provide the exact prompts used for the LLM-only baselines. These baselines evaluate the model's ability to answer complex queries relying solely on its pre-trained parametric knowledge, without access to the knowledge graph.

**Direct Prompting (Zero-shot).** Table 11 presents the prompt for the *Direct* setting. In this configuration, the model is instructed to provide the final answer immediately. This serves as a baseline for the model's raw internal knowledge retrieval capabilities.

**Few-shot Chain-of-Thought (CoT).** To establish a strong baseline, we implement the *Few-shot Chain-of-Thought* setting as shown in Table 12. We curate six diverse multi-hop QA exemplars (covering geography, art, history, and science) to enable in-context learning. This prompt structure is designed to elicit the model's inherent reasoning capabilities, ensuring that any performance gap observed in INSES can be attributed to the effective integration of graph-based evidence rather than a lack of reasoning effort in the baseline.

*Table 10.* Search with similarity expansion

| Iteration | The relevant status of each iteration |
|---|---|
| Query | Who was the spouse of a leading speaker against slavery and publisher of an antislavery newspaper? |
| Entities | ['leading speaker against slavery', 'antislavery newspaper', 'spouse', 'publisher'] |
| $V_{init}$ | ['Opponent of slavery', 'Anti-slavery newspaper', 'Husband and wife', 'Newspaper publisher'] |
| iter=0 | $V_{current}$: ['Opponent of slavery', 'Anti-slavery newspaper', 'Husband and wife', 'Newspaper publisher']. $T_{new\_selected}$: ['Thomas spottswood hinde → Occupation → Opponent of slavery', 'The north star → Is → Anti-slavery newspaper', 'Enos bronson → Was → Newspaper publisher']. $V_{candidate}$: ['Thomas spottswood hinde', 'The north star', 'Enos bronson']. $V_{sim}$: ['Pro-slavery southerner', 'Liberty party paper', 'Husbands and wives', 'Newspaper of record'] |
| iter=1 | $V_{current}$: ['Thomas spottswood hinde', 'The north star', 'Enos bronson', 'Pro-slavery southerner', 'Liberty party paper', 'Husbands and wives', 'Newspaper of record']. $T_{new\_selected}$: ['The north star → Published by → Frederick douglass']. $V_{candidate}$: ['Frederick douglass']. $V_{sim}$: ['Newspaper editor', 'The toronto star', 'Opponent of slavery', 'Federalist party', 'Husband and wife', "Country's newspaper of record"] |
| iter=2 | $V_{current}$: ['*Frederick douglass*', 'Newspaper editor', 'The toronto star', 'Federalist party', "Country's newspaper of record"]. $T_{new\_selected}$: ['The north star → Published by → Frederick douglass']. $V_{candidate}$: [ ]. $V_{sim}$: ['*Frederick douglass memorial and historical association*', 'Weekly newspaper', 'Federalists', 'Newspaper of record'] |
| iter=3 | $V_{current}$: ['*Frederick douglass memorial and historical association*', 'Weekly newspaper', 'Federalists']. $T_{new\_selected}$: ['*Helen pitts douglass → Created → Frederick douglass memorial and historical association*']. $V_{candidate}$: ['*Helen pitts douglass*']. $V_{sim}$: ['Frederick douglass', 'English language weekly newspaper', 'Federalist party'] |
| iter=4 | $V_{current}$: ['*Helen pitts douglass*', 'English language weekly newspaper']. $T_{new\_selected}$: ['*Helen pitts douglass → Is → Second wife of frederick douglass*']. $V_{candidate}$: ['*Second wife of frederick douglass*']. |
| Answer | Helen Pitts Douglass |

## G. Implementation Details of Naïve RAG

We implement Naïve RAG as both a strong baseline for standard retrieval tasks and the primary execution engine for non-complex queries within our routing pipeline. To ensure efficient and scalable retrieval, we utilize the Qdrant vector database as the storage backend and employ bge-base-en-v1.5 as the embedding model.

**Indexing Strategy.** For each dataset, we index the full corpus by embedding all available context passages. To maintain consistency with the original benchmarks, we preserve the original granularity of the text segments provided in the datasets, performing no additional splitting or merging.

**Inference and Confidence Estimation.** A critical feature of our Naïve RAG implementation is its ability to self-assess the reliability of its answers. As shown in the prompt template (Table 13), we instruct the LLM to output a structured JSON response containing three fields: `reasoning`, `answer`, and `confidence`.

- The `reasoning` field encourages Chain-of-Thought (CoT) processing to minimize hallucinations.

- The `confidence` score (ranging from 0 to 1) serves as the control signal for our router. If this score falls below a calibrated threshold, the system recognizes the query as ambiguous or complex, triggering the escalation to INSES for deeper graph-based reasoning.

*Table 11.* LLM only (Direct) Prompt

---

**LLM only (Direct) Prompt**

---

You are a helpful assistant that answers questions based on your own knowledge.

Question: {question}

Please provide your response in the following JSON format:

"answer": "Your final answer"

---

## H. Prompts for Automated Evaluation (LLM-as-a-Judge)

To overcome the limitations of rigid string-matching metrics (e.g., Exact Match), which often penalize correct but lexically distinct answers, we employ an LLM-as-a-judge paradigm. This approach is applied in two distinct experimental settings to ensure robust and semantically valid evaluation.

**Task 1: Semantic Equivalence Assessment.** In our main comparative experiments and ablation studies (Sections 4.5 and 4.6), the primary goal is to verify whether the model's generated answer matches the ground truth in meaning. As detailed in Table 14, the LLM judge is instructed to perform a semantic comparison robust to variations in phrasing, length, or synonym usage. The output is structured as a JSON object containing a boolean verdict and a brief justification, ensuring that the evaluation process is interpretable.

**Task 2: Evidence Sufficiency Verification.** In the robustness analysis on the MINE benchmark (Section 4.8), the evaluation objective shifts from answer correctness to graph quality assessment. Here, we need to verify whether the triples retrieved/constructed by different methods (KGGEN, GraphRAG, OpenIE) explicitly contain the target fact. As shown in Table 15, the LLM acts as a strict logical verifier. It checks if the Correct Answer is logically entailed by the provided Context (the retrieved triples), outputting a binary score (1 for success, 0 for failure) to quantify the recall capability of different graph construction methods.

*Table 12.* LLM only (Few-shot CoT) Prompt

---

**LLM only (Few-shot CoT) Prompt**

---

You are a helpful assistant that answers questions based on your own knowledge. Below are several examples of chain of thought. You can refer to these examples to think about the question and give the correct answer.

Your answer must be returned in JSON format with two fields: "reasoning" and "answer". The "reasoning" field should contain your step-by-step reasoning process, and the "answer" field should contain the final answer. The "answer" field should be as concise as possible and should not contain unnecessary explanations.

Examples of Chain of Thought:
Q: What language is primarily spoken in the country whose capital is Madrid?
A: First, the country whose capital is Madrid is Spain. Second, the primary language of Spain is Spanish. The answer is {Spanish}.
Q: Who painted The Starry Night and famously cut off part of his ear?
A: First, The Starry Night was painted by Vincent van Gogh. Second, the artist who cut off part of his ear is Vincent van Gogh. The answer is {Vincent van Gogh}.
Q: What continent contains the country whose capital is Nairobi?
A: First, Nairobi is the capital of Kenya. Second, Kenya is located in Africa. The answer is {Africa}.
Q: Which composer wrote The Magic Flute and was born in Salzburg?
A: First, The Magic Flute was composed by Wolfgang Amadeus Mozart. Second, Mozart was born in Salzburg. The answer is {Wolfgang Amadeus Mozart}.
Q: What element has the chemical symbol Fe and is used to make steel?
A: First, the chemical symbol Fe stands for iron. Second, iron is commonly used to make steel. The answer is {Iron}.
Q: Which planet is known as the Red Planet and has the volcano Olympus Mons?
A: First, the Red Planet is Mars. Second, Olympus Mons is a volcano on Mars. The answer is {Mars}.

Question: {question}

Please provide your response in the following JSON format:

"reasoning": "Your step-by-step reasoning process"
"answer": "Your final answer"

---

*Table 13.* Naïve RAG Prompt

---

**Naïve RAG Prompt**

---

You are a helpful assistant that provides accurate and concise answers based on the provided context.

Please answer the following query: {query}

Context information is below:
{context}

Your response must be in JSON format with three fields:
1. "reasoning": Your step-by-step reasoning process based on the context.
2. "answer": The final answer to the query, as concise as possible without unnecessary explanations.
3. "confidence": The confidence level of your answer, where 0 means no confidence and 1 means complete certainty. If you cannot derive a reasonable answer from the provided context, the returned confidence level should be low.

Example response format:
{
"reasoning": "Step 1: ... Step 2: ... Step 3: ...",
"answer": "Concise answer here",
"confidence": 0.8
}

JSON Response:

---

*Table 14.* Implementation Details of LLM as a judge in Section 4.5 and 4.6

---

**LLM as a judge Prompt**

---

You are an expert evaluator. Your task is to determine if the predicted answer is semantically equivalent to the ground truth answer for the given question.

Question: {question}
Ground Truth Answer: {ground_truth}
Predicted Answer: {prediction}

Instructions: - Compare the predicted answer and the ground truth answer in the context of the question.
- They are considered equivalent if they convey the same meaning, even if the wording is different.
- Respond in JSON format with two keys:
"is_equivalent": true or false,
"explanation": a brief explanation for your decision.

Example response:
{{
"is_equivalent": true,
"explanation": "Both answers correctly state that the capital of France is Paris."
}}

Important: Only output the JSON object and nothing else.

---

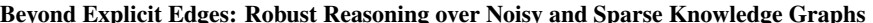

*Table 15.* Implementation Details of LLM as a judge in Section 4.8

**LLM as a judge Prompt**

You are an evaluator that checks if the Correct Answer can be deduced from the information in the Context."

Context:
{context}

Correct Answer:
{correct_answer}

Task: Determine whether the Context contains the information stated in the Correct Answer. Respond with "1" if yes, and "0" if no. Do not provide any explanation, just the number.

