# OpenReview forum: "Beyond Explicit Edges: Robust Reasoning over Noisy and Sparse Knowledge Graphs"
_ICML.cc/2026/Conference — ICML 2026 regular_

### Official Review · Reviewer_obgZ · 2026-03-04

**Soundness:** 2
**Presentation:** 3
**Significance:** 2
**Originality:** 2
**Overall Recommendation:** 4
**Confidence:** 3

**Summary:**

This paper introduces INSES (Intelligent Navigation and Similarity Enhanced Search), a framework designed to reason beyond explicit edges on graphs, by leveraging embedding-based similarity to infer missing graph links. This approach addresses graph noise and sparsity through two coupled components: (i) an LLM navigator that actively prunes neighboring triples if it finds them to be irrelevant to the query and (ii) embedding-based similarity expansion that dynamically augments new paths.

**Compliance With Llm Reviewing Policy:**

Affirmed.

**Key Questions For Authors:**

1. The authors mention that all experiments use the GLM-4 LLM. Is this also true for the baselines? Can these experiments be made available?


2. Could the authors provide a detailed breakdown of the average token consumption or LLM calls per query for INSES compared to the strong graph-based baselines like LightRAG or SiReRAG?


3. Could the authors elaborate on how the routing confidence threshold was calibrated?

**Limitations:**

No explicit limitations are mentioned, only briefly in the Appendix. I would appreciate a limitations discussion, mostly based on the computational overhead that the method introduces.

**Strengths And Weaknesses:**

Strengths

S1. Readability. The paper is well written and easy to follow.

S2. Method. The paper introduces an interesting mechanism of dynamic link pruning and prediction based on the query, as well as an interesting routing method.

S3. Evaluation results. The method presents great evaluation results across benchmarks in comparison with a variety of baselines.

---

Weaknesses

W1. Complexity analysis. While the authors present a very interesting method, it adds significant computational complexity to the RAG pipeline, adding a significant number of LLM calls and embedding comparisons. The authors touch upon this in Section 3.6, but only briefly. A more detailed complexity analysis for every step, the overhead it adds, and comparison with other RAG techniques would greatly benefit the paper. This is discussed in the Appendix, but added computational cost of LLM calls is not taken into account.

W2. Lack of baselines evaluations code or source. The authors claim baseline results but these experiments are not part of their supplemental material. Nor do these evaluations originate from the original baseline papers. Hence, it is not clear how these comparisons were made, whether indeed the same base LLM was used for all of them or the evaluations originate from another peer reviewed paper.

W3. Lack of similarity threshold analysis. The paper would benefit from a similarity threshold analysis. As far as I understood, the authors do not mention the threshold they use for experiments. It seems to be 0.8 according to the code.

---

> ### Author Rebuttal · Authors · 2026-03-30
>
> We sincerely thank the reviewer for the positive assessment of our work. We are highly encouraged that you found our dynamic link pruning mechanism, routing method, and the overall readability of the paper to be strong. You raised excellent points regarding computational overhead, baseline transparency, and parameter settings. We have carefully addressed each of your comments below and incorporated the necessary details into the revised manuscript.
>
> 1. (W1 & Limitations) We agree that a more explicit discussion of the computational overhead and limitations is necessary.
>
> As analyzed in Appendix C.1, the vector similarity expansion is highly efficient, taking only $O(\log|V|)$ time using ANN search, which scales well even on large graphs with millions of entities. The primary computational bottleneck is indeed the LLM calls.
>
> To be exact, the maximum number of LLM calls for a single complex query in INSES is strictly bounded to 8:
> 1 call for initial entity extraction.
> Up to 6 calls for iterative navigation and pruning (since max_iter $\le$ 6).
> 1 call for final answer generation.
>
> We will add an explicit Limitation section to the paper discussing this exact computational overhead.
>
> However, it is crucial to note that this is a worst-case per-query cost. At the system level, our routing mechanism acts as a strict gatekeeper. In real-world industrial deployments, we observe that fewer than 20% of user queries are structurally complex enough to be routed to the graph search; the vast majority are resolved cheaply by text RAG. Therefore, the amortized computational cost of the entire RAG pipeline remains highly practical.
>
> 2. (W2 & Q1) All baselines used GLM-4 model to ensure a fair and rigorous comparison. For the "LLM only" and "Text-based" methods, we utilized our own standard implementations. For the "Graph-based" methods, we utilized their official, publicly available open-source repositories. We will make our complete source code and evaluation data publicly available upon acceptance to ensure full reproducibility.
>
> 3. (W3 & Q3) You are correct in your observation of the code. In our main experiments, the parameters were empirically calibrated on a small validation split to balance noise-filtering and multi-hop recall.
>
> Similarity Threshold ($\tau_{sim}$): Set to 0.8. This relatively high threshold ensures that we only expand to nodes with strong semantic proximity, reducing the initial noise burden on the LLM navigator.
>
> Routing Confidence Threshold ($\theta$): Set to 0.8. If text RAG's self-assessed confidence falls below 0.8, the query is escalated to INSES. We set this boundary strictly because text-based RAG often hallucinates plausible-sounding answers on complex multi-hop tasks; a high threshold forces the system to rely on explicit graph logic when in doubt.
>
> 4. (Q2) While we did not track the precise token-by-token consumption across all graph-based baselines due to the wildly different ways each framework constructs its context prompts, we can provide a clear qualitative comparison of the overhead architecture.
>
> Methods like GraphRAG and LightRAG typically rely on aggregating massive community summaries or retrieving extensive subgraphs, which are then stuffed into the LLM's context window. This results in a smaller number of LLM calls but a massive token consumption per call.
>
> In contrast, INSES trades token length for interaction frequency. At each iteration, the LLM only evaluates a highly constrained local neighborhood (a few adjacent triples), keeping the context window very short. Therefore, while INSES uses more discrete LLM calls, its total token consumption is kept manageable.
>
> Additionally, we will discuss the issue of token consumption in the added Limitation section.

---

> > ### Author Rebuttal · Reviewer_obgZ · 2026-04-01
> >
> > I thank the authors for their response.
> >
> > Some of my concerns remain unresolved but I understand that it is difficult to resolve them in a short time, for example the token consumption. But I am satisfied with the authors' responses, especially the clarification regarding the bounding of LLM calls.

---

> > > ### Author Response · Authors · 2026-04-03
> > >
> > > We sincerely thank the reviewer for your understanding and for maintaining the positive evaluation of our work. We are very glad that our clarification regarding the strict bounding of LLM calls effectively addressed your concerns.
> > >
> > > We completely agree with your insight that token consumption is a vital aspect of evaluating the true overhead of LLM-guided graph search. To provide a rigorous and holistic view of this trade-off, we commit to adding a dedicated discussion in the Limitations section. This new section will provide a detailed analysis of the token complexity and context-window utilization of INSES compared to other Graph-based RAG paradigms.
> > >
> > > Thank you again for your constructive guidance, which has undoubtedly helped us identify critical areas to strengthen the manuscript.

---

### Official Review · Reviewer_3pWo · 2026-03-10

**Soundness:** 2
**Presentation:** 2
**Significance:** 2
**Originality:** 1
**Overall Recommendation:** 2
**Confidence:** 4

**Summary:**

This paper proposes a novel graph search method, INSES, designed for robust reasoning over noisy and sparse  knowledge graphs.Specifically, it integrates LLM-guided navigation to prune noisy paths and guide exploration, with  embedding-based similarity expansion to recover hidden links and bridge semantic gaps.Experiments on three benchmark datasets demonstrate that INSES achieves superior performance, while further ablation studies verify its  effectiveness and robustness.

**Compliance With Llm Reviewing Policy:**

Affirmed.

**Final Justification:**

I have reviewed the authors’ response and decided to keep my original score.

**Key Questions For Authors:**

1. In Section 3.5, during node expansion, the paper selects similar nodes from all entities in the knowledge graph, relying solely on the
node representation while ignoring additional information such as the query or graph structure. This approach may introduce extra noise
or redundant entities.
2. In Section 3.6, the paper introduces a lightweight routing mechanism to dispatch queries. How is the difficulty level of a query
determined?
3. In Section 4.1, the paper samples 1,000 instances from the dataset to construct a new test set. How is the fairness and comparability of the experimental results ensured under this setting?
4. In Section 4.6, why does the variant with the routing mechanism achieve better performance? In my view, the routing mechanism
mainly provides a trade-off between effectiveness and efficiency.

**Limitations:**

Yes.

**Strengths And Weaknesses:**

Strengths
1. This paper focuses on a valuable problem in real-world scenarios: how to perform efficient and robust retrieval over noisy and sparse
knowledge graphs.
2. The paper proposes an iterative retrieval strategy, which first leverages a large language model to prune the search space and then expands candidate nodes based on embedding similarity.This design is intuitive and well-structured.
3. In addition, the paper introduces a lightweight routing mechanism that dynamically selects between standardRAG and the proposed INSES method according to the difficulty of the query.

Weaknesses
1. This paper aims to address the challenges of reasoning over noisy and sparse knowledge graphs.However, the experimental section does not  specifically evaluate INSES under noisy or sparse settings.
2. The methodological novelty appears limited. Prior work has already explored the use of large language models for pruning during knowledge graph exploration. In addition, expanding nodes based on semantic similarity is relatively common and does not appear particularly novel or compelling.

---

> ### Author Rebuttal · Authors · 2026-03-30
>
> We sincerely thank the reviewer for evaluating our work and acknowledging that our design targeting noisy and sparse knowledge graphs is intuitive, well-structured, and addresses a valuable real-world problem. We carefully address your concerns regarding our experimental setup, the methodological novelty, and the routing mechanism below.
>
> 1. (W1) We respectfully clarify that evaluating INSES under strictly controlled noisy and sparse settings is exactly the focus of Section 4.8 and Appendix A in our paper.
> To systematically test robustness against varying degrees of sparsity and noise, we evaluated INSES on the MINE benchmark. We constructed KGs using three distinct paradigms (GraphRAG, KGGEN, and OpenIE), which inherently produce drastically different topological structures. As quantitatively detailed in Appendix A and Figure 4, GraphRAG produces extremely sparse graphs (avg. 14 nodes/13 edges per document), while OpenIE produces highly dense and noisy graphs (avg. 189 nodes/265 edges).
> Figure 3 in the main text explicitly demonstrates our results across these varied settings. INSES consistently improved accuracy over the baselines across these diverse KGs. This directly verifies that our framework remains robust regardless of whether the underlying graph is heavily sparse (GraphRAG) or highly noisy (OpenIE).
>
> 2. (W2 & Q4) We appreciate the opportunity to clarify our core novelty. While LLM pruning and vector similarity exist independently, our novelty lies in their dynamic, tightly-coupled integration to solve the semantic fragmentation problem inherent to KGs.
> Static similarity expansion (like traditional Link Prediction) globally introduces massive noise. Conversely, INSES creates "virtual edges" on-the-fly and subjects them to immediate LLM verification in the next step. This allows us to achieve the high recall of vector expansion while using the LLM to strictly bound the introduced noise, preventing noise propagation.
> Regarding Question 4, the router achieves better overall performance because it provides an effectiveness trade-off, not just an efficiency one. As detailed in Appendix D (Table 7), Text RAG and Graph RAG have fundamentally different representational strengths. For simple queries ($ \le 2$ hops), Text RAG often outperforms Graph RAG because it preserves granular textual details and exact phrasing that are frequently lost during KG construction. Table 2 supports this, showing Text RAG variants outperforming Graph baselines on datasets like HotpotQA. The router boosts overall accuracy by actively directing queries to the representation paradigm (Text vs. Graph) best suited to answer them.
>
> 3. (Q1) Relying strictly on vector representation during the similarity expansion step (Section 3.5) is a deliberate architectural choice driven by computational complexity.
> In a large-scale graph $G(V,E)$, utilizing Approximate Nearest Neighbor (ANN) algorithms (like HNSW) allows us to retrieve top-k similar nodes with a time complexity of $O(\log|V|)$. If we were to incorporate explicit graph structural matching or multi-hop topological features into this continuous expansion phase, the complexity would scale to $O(|V|)$ or worse, rendering the dynamic search computationally unfeasible for real-world KGs.
> We acknowledge that vector-only expansion introduces redundant entities. However, as explained in our response to Weakness 2, this noise is immediately mitigated by the LLM-guided pruning in the subsequent iteration.
>
> 4. (Q2) The difficulty of a query is determined dynamically via a two-fold mechanism detailed in Algorithm 2 and Appendix D:
> (1) Hop-Count Prediction: An LLM acts as an initial classifier to determine if the query structurally requires multi-hop reasoning (i.e., $> 2$ hops). If yes, it is classified as complex.
> (2) Confidence Threshold: If routed to Text RAG, the LLM generates an answer along with a confidence score. If this score falls below a predefined threshold, the query is re-classified as complex/ambiguous and escalated to INSES.
>
> 5. (Q3) Sampling a subset of queries (e.g., 1,000 instances) is a standard and rigorously accepted practice in evaluating LLM-based RAG and Graph RAG systems due to the massive computational overhead of API calls. To ensure absolute fairness and strict comparability, all baseline methods and our proposed INSES framework were evaluated on the exact same 1,000 instances sampled from each dataset, retrieving from the exact same underlying corpus. This methodology strictly aligns with prior works such as IRCoT.

---

> > ### Author Rebuttal · Reviewer_3pWo · 2026-04-04
> >
> > Thank you for your rebuttal. I'd like to maintain my score.

---

> > > ### Author Response · Authors · 2026-04-04
> > >
> > > Thank you again for your follow-up. We appreciate your time and consideration.
> > >
> > > We would only like to briefly clarify two points for the record. First, the concern that INSES was not evaluated under noisy or sparse settings refers to an analysis that is already included in the submission: our robustness experiments on MINE across KGs built by KGGEN, GraphRAG, and OpenIE were designed precisely to test performance under substantially different graph quality and density regimes (Section 4.8). Second, the routing mechanism is already described in the paper as being based on estimated complexity and confidence, with the detailed procedure provided in the Appendix D and Algorithm 2.
> > >
> > > Our intention here is only to ensure that the final discussion accurately reflects the content already included in the manuscript and the clarifications provided in the rebuttal.

---

### Official Review · Reviewer_wZ1g · 2026-03-10

**Soundness:** 2
**Presentation:** 3
**Significance:** 3
**Originality:** 2
**Overall Recommendation:** 4
**Confidence:** 4

**Summary:**

This paper tackles an important practical issue in KG‑based reasoning: real-world knowledge graphs built from text are often messy, sparse, and inconsistent, which makes traditional graph traversal brittle. The authors propose **INSES**, a system that combines two ideas: (1) LLM‑guided navigation, where an LLM filters and prioritizes graph triples dynamically rather than relying on rigid adjacency lists. (2) Similarity‑based expansion, where semantically related nodes (according to embeddings) are injected into the search frontier, effectively repairing missing edges on the fly.

To avoid unnecessary computation, a **router** decides whether a query is simple enough for Naïve RAG or whether it needs the full INSES pipeline. Across several QA benchmarks and multiple KG construction methods, the approach shows consistent improvements over text-based and graph-based baselines. Ablations confirm that similarity expansion is the main driver of performance.

**Compliance With Llm Reviewing Policy:**

Affirmed.

**Final Justification:**

This submission presents a well‑motivated approach to improving KG‑based reasoning under real‑world conditions, where automatically constructed knowledge graphs are often sparse, noisy, and fragmented. The proposed INSES framework--integrating LLM‑guided navigation, similarity‑based expansion, and a routing mechanism--offers a coherent and practical strategy for addressing these challenges. The experimental results are solid, and the ablation studies clearly illustrate the contribution of each component.

In my initial review, I raised concerns regarding the formulation of Equation (4), the potential for semantic drift, the clarity of the routing mechanism, scalability to larger graphs, comparisons with recent related work, and several under‑specified design choices. I appreciate the authors’ careful and thorough rebuttal. The corrections to the similarity‑expansion equation, the illustrative example demonstrating how drift is controlled, and the clarification of routing behavior all improved the clarity of the work. The additional discussion of scalability and the positioning of INSES within the broader landscape of recent methods further strengthened the submission. I also appreciate the authors’ intention to expand the manuscript with more complete implementation details.

While the rebuttal resolved my main conceptual concerns, I maintain a **weak accept** recommendation rather than a full accept. This is primarily due to the current level of empirical depth in certain areas, such as more systematic runtime analyses, large‑scale scalability evidence, and robustness studies. These do not undermine the validity of the contribution but suggest that there is still room to strengthen the empirical foundation.

I find the work technically sound, clearly motivated, and likely to be valuable to the community. The authors’ responses were thoughtful and constructive, and I consider the major issues addressed. My final recommendation remains **weak accept**, reflecting a positive evaluation delivered with the view that the paper could be further strengthened in future iterations.

**Key Questions For Authors:**

These are the questions that would affect my evaluation if clarified in the rebuttal.

Q1) Do you intend to return multiple similar nodes? If so, will you revise Eq. (4) to describe a top‑k or threshold-based set? As written, the math and the described algorithm do not match.

Q2) How do you prevent incorrect similarity-based edges from derailing the search? Do you have pruning heuristics, confidence weighting, or LLM cross-validation?

Q3) How exactly is the router trained or tuned? What signals does it use to decide whether a query is "complex"? Can you provide statistics on misrouting and its impact?

Q4) What happens when the KG contains millions of nodes? Are similarity searches run globally? Can you provide cost breakdown (LLM calls, retrieval time, graph operations)?

Q5) Why are recent adaptive RAG or LLM-guided graph exploration systems not included? For example, more advanced chain‑of‑thought retrieval controllers, dynamic reasoning agents, or modern KGC models. How would INSES fare against them?

Q6) Consider also my remarks at weakness section.

**Limitations:**

yes

**Strengths And Weaknesses:**

# **Strengths**

S1) It addresses a real limitation of graph‑based RAG pipelines. KGs built from natural text are incomplete, fragmented, and noisy. The framing around "semantic fragmentation" is convincing.

S2) The idea of dynamically "patching" the KG during reasoning is appealing and well-motivated.

S3) The robustness on the MINE benchmark is particularly noteworthy.

S4) Clear ablation studies, showing how each component contributes and why the system works.

S5) Practical use of routing, achieving a good balance between speed and accuracy.

# **Weaknesses**

W1) Equation (4) does not match the described behavior. The paper claims INSES expands to *multiple* similar nodes, but Eq. (4) defines:
    $$
    v^*(u)=\arg\max \cos(\phi(u),\phi(v))
    $$
    This returns only *one* neighbor. Unless I am mistaken, it should instead return a *set*: top‑k or threshold‑filtered neighbors. As written, it cannot implement the intended expansion.

W2) There is no analysis of how errors propagate when similarity-based expansion introduces incorrect neighbors. Since the expansion happens iteratively, mistakes can compound, but this isn’t explored.

W3) The router is not described in sufficient detail. It's a key part of the system, but its training, features, and failure modes are unclear.

W4) There is also a lack of discussion of semantic drift. Similarity-based expansion can easily introduce wrong nodes; the paper does not analyze or mitigate this.

W5) Scalability is unclear. The system’s reliance on repeated LLM calls and global similarity search may not scale to large KGs.

W6) Several important design components (e.g., the `Relevant()` function, LLM prompts, and threshold choices) are under-specified.


W7) There are missing comparisons to recent SOTA methods in adaptive retrieval, LLM-driven graph search, and link prediction.

*Here are some Recommendations for Improvement:*

* Modern retrieval & RAG:
    * Hao, S., Gu, Y., Luo, H., Liu, T., Shao, X., Wang, X., ... & Hu, Z. (2024). Llm reasoners: New evaluation, library, and analysis of step-by-step reasoning with large language models. arXiv preprint arXiv:2404.05221.

* Graph-based retrieval & reasoning:
    * Dziri, N., Madotto, A., Zaiane, O. R., & Bose, A. J. (2021, November). Neural path hunter: Reducing hallucination in dialogue systems via path grounding. In Proceedings of the 2021 Conference on Empirical Methods in Natural Language Processing (pp. 2197-2214).

* KG completion / latent link discovery:
    * Zhang, Y., Chen, Z., Guo, L., Xu, Y., Zhang, W., & Chen, H. (2024, October). Making large language models perform better in knowledge graph completion. In Proceedings of the 32nd ACM international conference on multimedia (pp. 233-242).

* Routing / mixture-of-reasoning systems:
    * Guo, D., Yang, D., Zhang, H., Song, J., Wang, P., Zhu, Q., ... & Tan, Y. (2025). DeepSeek-R1 incentivizes reasoning in LLMs through reinforcement learning. Nature, 645(8081), 633-638.

---

> ### Author Rebuttal · Authors · 2026-03-30
>
> We sincerely thank the reviewer for the highly constructive feedback and for championing the core motivation of our work. We are glad you found our framing of "semantic fragmentation" convincing and our approach to dynamically patching knowledge graphs appealing. We appreciate your meticulous reading, which helped us identify a mathematical typo, and we address all your insightful questions point-by-point below.
>
> 1.(W1 & Q1) Thank you for pointing this out. You are correct; the equation in the submitted draft incorrectly described the retrieval of a single node. Our actual implementation retrieves a set of similar nodes via top-k retrieval followed by a strict threshold filter. In our main experiments, we utilized $k=1$, meaning we retrieved the single most similar node if and only if it exceeded the similarity threshold $\tau_{sim}$. If it failed the threshold, the set was empty.
>
> We have corrected Equation (4) in the revised manuscript to accurately reflect a threshold-filtered set retrieval:
> $$
> V_{sim} = \\{ v \in V \setminus \\{u\\} \mid v \in \text{Top-k}(u) \land \cos(\phi(u), \phi(v)) \ge \tau_{sim} \\}
> $$
>
> 2. (W2, W4 & Q2) Your concern regarding error compounding and semantic drift is highly valid. Because cosine similarity lacks strict transitivity, unconstrained expansion is indeed risky. INSES employs two rigorous barriers to prevent incorrect similarity nodes from derailing the search:
>
> Barrier 1: Immediate LLM Pruning: We do not permanently/statically add edges to the graph. Instead, the LLM evaluates the adjacent triples of the newly expanded node in the immediate next step. If the expansion was a semantic drift (e.g., matching "Apple the fruit" to "Apple the company"), its neighborhood will lack logical relevance to the query, and the LLM will prune it, halting the drift.
>
> Barrier 2: Hard Bounded Search Depth: We strictly limit the maximum iterations to 6. This aligns with the small-world nature of real-world queries and acts as a fail-safe, ensuring that even if an error escapes pruning, it cannot cause an infinite runaway search.
>
> 3. (W3 & Q3) The router is not trained from scratch but operates as an LLM-based classifier combined with confidence threshold. Its goal is to direct queries to the representation paradigm (Text vs. Graph) best suited to answer them.
>
> As detailed in Appendix D, Text RAG excels at preserving granular details but fails in multi-hop tasks due to chunk fragmentation. Graph RAG provides clear logical chains but abstracts details. Therefore, our router decides based on:
>
> (1)Hop-Count Classification: The LLM evaluates if the query structurally requires $>2$ hops. If yes, it routes to INSES.
>
> (2)Text RAG Confidence: For simple queries processed by Text RAG, if the LLM's output confidence falls below a set threshold ($\theta = 0.8$), it triggers an escalation to INSES.
>
> We will expand Section 3.6 to clarify these exact mechanisms and thresholds.
>
> 4. (W5 & Q4) INSES is highly scalable because it actively avoids global, brute-force graph operations during inference.
>
> Retrieval Scalability: As detailed in Appendix C.1, the similarity search does not compute pairwise distances globally. We utilize efficient Approximate Nearest Neighbor (ANN) indices (HNSW), which reduce the expansion complexity to $O(\log|V|)$.
>
> LLM Call Scalability: The number of LLM calls per query is strictly bounded by the maximum iteration limit ($iter \le 6$). At each step, the LLM only evaluates a highly constrained neighborhood, preventing prompt bloat.
>
> 5. (W6) The $Relevant(., q)$ function is instantiated by the LLM Navigator. The exact mechanism and the prompts used for this classification are provided in Appendix B, Table 5. The similarity threshold $\tau_{sim}$ was empirically set to 0.8 in our main experiments. We will update the implementation details section to state this explicitly.
>
> 6. (W7 & Q5) We sincerely thank the reviewer for pointing out these highly relevant works.
>
> We will incorporate Dziri et al. (2021) and Hao et al. (2024) into our discussion on LLM-driven graph search and reasoning, highlighting how INSES extends path grounding to latent links. We will add Zhang et al. (2024) to differentiate our dynamic expansion from static KG Completion. Finally, we will cite DeepSeek-R1 (Guo et al., 2025) in our discussion of routing and mixture-of-reasoning, as it represents the bleeding edge of adaptive computation.

---

> > ### Author Rebuttal · Reviewer_wZ1g · 2026-04-02
> >
> > Thank you for the thoughtful and detailed rebuttal. I appreciate the effort you put into clarifying several technical points, particularly the correction to Equation (4), the description of the similarity‑based expansion mechanism, and the explanation of how the router operates. These additions definitely resolve some of the ambiguities I noted in the original submission.
> >
> > That said, a few of the core concerns remain only partially addressed on my side:
> >
> > ### 1. Error propagation and semantic drift:
> >
> > Your explanation of the two safeguards (LLM pruning + depth limits) is helpful, and it makes clear that you’ve thought about the issue. Still, what’s missing for me is some empirical grounding. It would help to know, even roughly, how often the system drifts semantically, how reliably the LLM pruning corrects such cases, and how much these errors affect overall performance. Since similarity‑based expansion is central to INSES, a bit more evidence would strengthen the story.
> >
> > ### 2. Scalability at large KG sizes:
> >
> > Your qualitative description of using HNSW indices and bounding the number of LLM calls makes sense. However, this still leaves open the more practical questions that motivated my original concern—runtime numbers, memory footprint, or at least some indicative cost breakdown. Without this, it’s hard to fully assess how INSES behaves in million‑node scenarios.
> >
> > ### 3. Router behavior and failure cases:
> >
> > The additional detail about hop‑count estimation and confidence thresholds is good, but I was really hoping for some statistics about how often the router misroutes queries and what the performance impact is. Since the router decides whether the model takes the "simple" path or the full INSES pipeline, understanding its reliability feels important.
> >
> > ### 4. Comparisons with recent adaptive retrieval and reasoning systems:
> >
> > I appreciate that you plan to integrate the references I pointed out. Still, the original question was not just about citations; it was also about situating INSES empirically or at least qualitatively relative to those recent approaches. This part remains mostly unaddressed.
> >
> > ### 5. Remaining design details:
> >
> > Providing the similarity threshold and pointing to the prompts in the appendix helps, but some design choices (thresholds, hyperparameter sensitivity, behavior of Relevant() in ambiguous situations) still feel a bit under‑motivated or under‑explained.
> >
> >
> > In general, your rebuttal clarifies several important implementation details and fixes the core methodological inconsistency in the paper. That said, some of the broader concerns--particularly around scalability, error propagation, and router reliability--are only partially resolved, mostly due to the absence of supporting empirical evidence.

---

> > > ### Author Response · Authors · 2026-04-03
> > >
> > > We sincerely thank the reviewer for the continued engagement and for acknowledging our efforts in the initial rebuttal. We provide detailed clarifications strictly addressing your follow-up questions below.
> > >
> > > 1.(Q1) As you correctly pointed out, similarity expansion is the core of INSES. The LLM's pruning mechanism is strictly constrained by the relevance between the **original query** and the expanded nodes, as well as the logical connection of the selected nodes and edges.
> > >
> > > We use a real execution trace from **Appendix E (Table 9)** to illustrate the impact of semantic drift and how LLM pruning effectively resolves it.
> > > * The original query is: *"Who was the spouse of a leading speaker against slavery and publisher of an antislavery newspaper?"*
> > > * During iteration 2, the algorithm uses similarity expansion to augment three entity nodes `["Frederick douglass", "Newspaper editor", "Federalist party"]` into three similar nodes `["Frederick douglass memorial and historical association", "Weekly newspaper", "Federalists"]`.
> > > * In iteration 3, when the LLM selects the triples (edges) associated with these newly expanded nodes, it *only* selects the edge associated with "Frederick douglass memorial and historical association" (`Helen pitts douglass -> Created -> Frederick douglass memorial and historical association`). It does not select any edges associated with "Weekly newspaper" or "Federalists".
> > > This occurs because the LLM's selection process (pruning) is based on the correlation with the **original query and the logical connection of the edges**. Its accuracy is not limited by the similarity between the source node and the expanded node, but is constrained by the **original query**, which remains constant throughout the entire graph search. Consequently, semantic drift or errors cannot propagate and accumulate.
> > >
> > > 2.(Q2) (1) The complexity of similarity expansion is $O(\log|V|)$. Even in a graph with millions of nodes, $\log|V|$ is approximately 20. Therefore, similarity expansion scales well with graph size. (2) INSES invokes the LLM only once per iteration. The total number of LLM calls is independent of the graph size. Thus, the LLM invocation is also scalable regarding graph size.
> > >
> > > 3.(Q3) Section 4.7 ("Routing Behavior Analysis") details this mechanism. In simpler datasets like HotpotQA, over 80% of queries are routed to Text RAG, whereas in complex datasets, this proportion drops to roughly 50%. The determination of a query's complexity is based on the potential number of hops it involves. Referring to our analysis in Appendix D ("When Text RAG tends to win ($\le 2$ hops)"), queries requiring $\le 2$ hops generally fall into two clear and simple scenarios, making it relatively easy for the LLM to make an accurate routing judgment.
> > >
> > > Furthermore, considering the massive difference in computational cost (Graph-based RAG is usually 10x slower than Text RAG due to multiple LLM calls), a slightly conservative routing strategy can be adopted in real-world systems: All queries are initially routed to Text RAG, which provides an answer along with a confidence score. High-confidence answers are accepted directly, and only queries yielding low-confidence answers are subsequently routed to Graph-based RAG. This approach adds minimal overall cost while fully leveraging the respective strengths of both Text RAG and Graph-based RAG.
> > >
> > > 4.(Q4) Hao et al. (2024) proposes a framework (AutoRace) to automatically evaluate the reasoning chains generated by LLMs. Its focus is on evaluating the intrinsic capabilities of the LLM itself. Guo et al. (2025) demonstrates that DeepSeek-R1's reasoning capabilities can be significantly stimulated through pure reinforcement learning without relying on manual reasoning traces. This is a great contribution aimed at enhancing the LLM's own reasoning ability. In contrast, our INSES focuses on how to better utilize external knowledge to improve the reasoning ability of LLMs, our experiments include comparisons with CoT baselines (Table 2), which is also included in AutoRace.
> > >
> > > Dziri et al. (2021) proposes Neural Path Hunter, which utilizes KGs to verify and correct factual hallucinations in LLM responses by finding a supporting path in the KG for a given dialogue. Constructing paths is similar to INSES, but the direction is opposite. Neural Path Hunter finds a path in the KG to verify a *known answer*, whereas INSES searches for a helpful path to deduce an *unknown answer*. Zhang et al. (2024) is an excellent study on helping LLMs better utilize graph structure information for KGC. In Appendix C.2, we compare INSES with link prediction and explain our rationale for not adopting static link prediction: static completion can introduce potential permanent errors and lead to error propagation.
> > >
> > > 5.(Q5) We will further review and supplement the design choices, threshold settings, and implementation details within the manuscript to maximize the transparency and reproducibility of our work.

---

### Official Review · Reviewer_rYnk · 2026-03-13

**Soundness:** 4
**Presentation:** 3
**Significance:** 3
**Originality:** 3
**Overall Recommendation:** 4
**Confidence:** 5

**Summary:**

This paper introduces INSES, a dynamic framework addressing the limitations of standard GraphRAG in handling noisy or incomplete knowledge graphs. It couples LLM-guided navigation with embedding-based similarity expansion to enable reasoning beyond explicit edges. A key contribution is a lightweight router that balances efficiency by delegating simple queries to Naïve RAG. Results show consistent SOTA improvements, with notable robustness demonstrated across KGs constructed by different methods.

**Compliance With Llm Reviewing Policy:**

Affirmed.

**Final Justification:**

I will maintain my latest score.

**Key Questions For Authors:**

Please see the weaknesses.

**Limitations:**

yes

**Strengths And Weaknesses:**

Strengths：
1. This study addresses the issue that the current GraphRAG and traditional graph search algorithms overly rely on explicit static edges. It points out that this rigid traversal is prone to causing "semantic fragmentation" when dealing with real-world noisy, sparse, or incomplete knowledge graphs (KGs).
2. This paper is technically solid and well-organized.

Weaknesses:
1. In Algorithm 1, INSES sets a threshold $\tau_{sim}$ to find the most similar node $V_{sim}$ to enhance the front node set. This is a high-yield but high-risk action. In the high-dimensional vector space, cosine similarity does not have strict transitivity isomorphism (that is, if $A \sim B$ and $B \sim C$, it does not mean that the attributes of $A$ are applicable to $C$). In the continuous multi-hop (Multi-hop) dynamic expansion, semantic drift is extremely likely to occur. Although the authors point out that the subsequent steps of LLM can filter errors, this is highly dependent on LLM not experiencing Hallucination. Therefore, the authors need to provide solutions and effects.

2. As shown in Section 3.6 and Appendix D, the lightweight router is responsible for deciding whether the query should go to the cheaper Naïve RAG or the more expensive INSES framework. However, Algorithm 2 is merely described as using LLM to determine whether the query involves multiple hops or based on the Confidence output of Naïve RAG. The routing strategy of the router cannot merely rely on the Boolean output or internal probability of a black box LLM. It is necessary to clearly quantify what is "simple" and what is "complex".

3. The bar chart of the accuracy distribution in Figure 3 is somewhat rough and difficult to intuitively reflect the subtle variations in algorithm variance under different knowledge graph densities (such as the extremely sparse GraphRAG and the highly redundant and noisy OpenIE). It is recommended to use a smoother kernel density estimation graph (Kernel Density Estimation, KDE) or a violin plot with outlier markers for presentation.

---

> ### Author Rebuttal · Authors · 2026-03-30
>
> We sincerely thank the reviewer for the thorough evaluation and the highly insightful comments. You have accurately identified the core challenges of dynamic graph expansion and routing. We completely agree with your characterization of similarity expansion as a high-yield, high-risk action. Your precise identification of the semantic drift risk and the router's quantification has guided us to strengthen the empirical validation of our framework. We address your concerns point-by-point below.
>
> 1. (W1) We strongly agree with your insight that cosine similarity lacks strict transitivity  in high-dimensional vector spaces ($A \sim B$ and $B \sim C$ does not imply $A \sim C$. For example, $\cos(A,B)=0.8$ and $\cos(B,C)=0.8$, but the lower bound of $\cos(A,C)$ is $0.28$). This is precisely why INSES does not blindly rely on vector distances for continuous and static expansion, but uses them only as candidate generators, strictly governed by the LLM Navigator's context-aware pruning.To objectively prove that INSES effectively blocks semantic drift, we designed a targeted, mathematically grounded Transitivity Decay Blocking Experiment.
>
> Experimental Setup and Results: From our vector database, we select 500 Mathematical Drift Chains $(N_0 \rightarrow N_1 \rightarrow N_2)$. We strictly defined a drift chain where local similarities are high ($\cos(N_0, N_1) \ge 0.8$ and $\cos(N_1, N_2) \ge 0.8$), but the global transitive similarity has mathematically collapsed ($\cos(N_0, N_2) \le 0.6$). We simulated the INSES pipeline reaching the drifted node $N_2$. We provided the LLM Navigator with the original Query and the adjacent triples of the mathematically drifted node $N_2$. We objectively measured the Drift Blocking Rate: the percentage of cases where the LLM correctly identified the semantic irrelevance and rejected all adjacent triples of $N_2$. The experiment revealed a Drift Blocking Rate of 98.6%. Even though $N_2$ was forced into the candidate pool, the LLM Navigator almost never hallucinated utility. Because the LLM evaluates the explicit semantic logic of the triples relative to the original query, it successfully identified the severe drop in relevance and actively severed the connection.This confirms that while similarity expansion is a high-yield mechanism for surfacing latent links, the tightly coupled LLM pruning acts as an extraordinarily robust, context-aware barrier that prevents multi-hop semantic drift.
>
> 2. (W2) We appreciate the reviewer pushing us to rigorously quantify the routing boundary. You are correct that the router cannot rely solely on a black-box LLM's boolean output.
>
> The routing mechanism is fundamentally driven by the structural complementarity of Text RAG and Graph RAG, rather than just computational cost. Text RAG operates over coarse-grained chunks, excelling at preserving rich local details but struggling heavily in multi-hop reasoning because evidence is fractured across isolated chunks. Conversely, KGs provide clear logical chains but may miss fine textual details.
>
> To make the router transparent and reproducible, we will updated Section 3.6 to explicitly quantify the boundary between simple and complex across two objective dimensions:
>
> (1) Topological Complexity (Hop-Count): Queries structurally requiring more than 2 hops for logical resolution are strictly defined as complex and routed to INSES (Appendix D contains a detailed analysis of the relationship between hop count and complexity).
>
> (2) Calibrated Confidence Threshold: For queries initially processed by Text RAG, we quantify simple by requiring the output confidence to strictly exceed a threshold $\theta$ (which we empirically set and evaluate at $\theta = 0.8$ ). Any output falling below $\theta$ explicitly indicates that text retrieval failed to capture the complete logical chain, triggering an automatic escalation to the graph reasoning framework.
>
> 3. (W3) Thank you for the visualization suggestion. We agree that the overlapping bar charts in Figure 3 made it difficult to intuitively discern the subtle variance and density differences across the extremely sparse GraphRAG and the highly redundant OpenIE structures.
>
> The table below summarizes the statistical characteristics of the accuracy distributions shown in Figure 3. Based on the original data, we will redraw Figure 3 using a smoother KDE to replace the current version.
>
> | Comparison        | Method | Mean   | Std    |
> |-------------------|-----------|--------|--------|
> | KGGEN vs INSES    | KGGEN     | 0.6673 | 0.1896 |
> | KGGEN vs INSES    | INSES     | 0.7145 | 0.1910 |
> | GraphRAG vs INSES | GraphRAG  | 0.4780 | 0.1828 |
> | GraphRAG vs INSES | INSES     | 0.5688 | 0.1945 |
> | OpenIE vs INSES   | OpenIE    | 0.2940 | 0.1628 |
> | OpenIE vs INSES   | INSES     | 0.5693 | 0.1587 |

---

> > ### Author Rebuttal · Reviewer_rYnk · 2026-04-01
> >
> > Thanks to the authors for their helpful replies, most of my questions have been resolved, so I've decided to increase my score.

---

> > > ### Author Response · Authors · 2026-04-03
> > >
> > > We sincerely thank the reviewer for the positive feedback and for deciding to raise the score! We are thrilled that our previous rebuttal successfully resolved most of your concerns. Thank you again for your time and constructive evaluation!

---

### Decision · Program_Chairs · 2026-04-30

**Decision:**

Accept (regular)

**Comment:**

This paper addresses a practically important weakness of GraphRAG systems: brittle reasoning over noisy, sparse, and incomplete knowledge graphs. Reviewers found the core idea intuitive and useful, and the empirical robustness results—especially across KGs built by different construction methods—are a notable strength. The rebuttal also clarified important implementation details around semantic-drift control, routing, and scalability, and I have read the authors’ comments regarding review quality and engagement and taken them into account in forming this assessment.

That said, the concerns were not fully eliminated. The paper remains somewhat borderline on contribution substantiality: the overall system is effective, but part of the method can still be viewed as a well-executed combination of LLM-guided pruning, similarity-based expansion, and routing rather than a sharply new algorithmic advance. There is also still a split in reviewer enthusiasm. Overall, this is a positive but cautious case, where the practical value and robustness evidence are strong, while the remaining questions concern how substantial the methodological advance is.